# Plasmonic Ni-doped $W_{18}O_{49}$ with dual active sites drives efficient methanol dehydration to dimethyl ether

Dehua Tian[1], Yinlan Liang[1], Zhaoke Zheng [2]✉, Liang Mao[3], Xiaoyan Cai[3], Yizhen Chen[4], Xiangxian Wang[4], Xiaolei Liu[1]✉, Juan Li[1], Zeyan Wang[2], Can Xue[5], Baojun Li[1] & Zaizhu Lou[1]✉

Photocatalytic methanol dehydration to dimethyl ether (DME) offers a sustainable alternative to energy-intensive thermocatalysis, yet its practical application remains constrained by low efficiency. Herein, we designed Ni-doped plasmonic $W_{18}O_{49}$ nanowires that synergistically integrates low-coordinated W and Ni dual active sites with surface plasmon resonance for enhanced photocatalytic performance. The synergistic effect of W and Ni dual sites is amplified by plasmonic electron oscillations to facilitate the C-O bond cleavage and C-O-C coupling, driving efficient methanol-to-DME conversion. The optimized $Ni_{0.66}$-$W_{18}O_{49}$ achieves a DME yield of $133.7 \pm 3.3$ mmol $g^{-1}$ $h^{-1}$ with 98.7% selectivity under 400 mW $cm^{-2}$ illumination. The versatility of the catalyst is demonstrated through $C_{2+}$ alcohol dehydration, achieving 40-80% rate enhancements and a recorded isobutylene yield of 3.7 mol $g^{-1}$ $h^{-1}$. This study highlights the huge potential of rationally engineered plasmonic semiconductors in solar-driven chemical synthesis, particularly for C-O bond activation and coupling reactions.

Dimethyl ether (DME) is extensively applied as a clean fuel, friendly aerosol, and green refrigerant[1–3]. According to the latest market analysis, the global DME market was valued at approximately USD 8.69 billion in 2023 and is expected to grow to USD 25.4 billion by 2036[4], reflecting a significant growth potential. Conventional industrial production relies on energy-intensive methanol dehydration over metal-supported aluminosilicate catalysts at elevated temperatures (230–400 °C) and pressures (0.5–1.5 MPa)[5–7]. This thermodynamic demand highlights the critical need for catalyst innovations enabling efficient methanol conversion under mild conditions. While solar-driven photocatalysis presents a sustainable alternative, current catalysts exhibit insufficient activity (DME generation rate of 4.3 mmol $g^{-1}$ $h^{-1}$) for industrial implementation[8–10]. Plasmonic photocatalysis offers a superior-active approach for chemical transformation by leveraging surface plasmon resonance (SPR) to concentrate optical energy at catalytically active interfaces[11–15]. Our group previously demonstrated the exceptional capability of plasmonic $W_{18}O_{49}$ nanowires in alcohol-to-alkene conversion, achieving an isobutylene production rate of 1.8 mol $g^{-1}$ $h^{-1}$ through plasmon-driven C-O bond activation[16–18].

Unlike the intramolecular dehydration for alkene generation, DME synthesis necessitates a bifunctional mechanism involving sequential C-O bond cleavage and C-O-C coupling, and this process requires spatially distinct active sites[19]. While pristine $W_{18}O_{49}$ effectively facilitates C-O bond dissociation, its inability to mediate C-O-C coupling restricts DME productivity. Systematic density functional theory (DFT) simulations (Fig. 1) reveal atomic-level insights into

[1]Guangdong Provincial Key Laboratory of Nanophotonic Manipulation, Institute of Nanophotonics, College of Physics & Optoelectronic Engineering, Jinan University, Guangzhou, China. [2]State Key Laboratory of Crystal Materials, Shandong University, Jinan, China. [3]School of Materials Science and Physics, China University of Mining and Technology, Xuzhou, Jiangsu Province, China. [4]School of Science, Lanzhou University of Technology, Lanzhou, China. [5]School of Materials Science & Engineering, Nanyang Technological University, Singapore, Singapore. ✉e-mail: zkzheng@sdu.edu.cn; liuxiaolei@jnu.edu.cn; zzlou@jnu.edu.cn

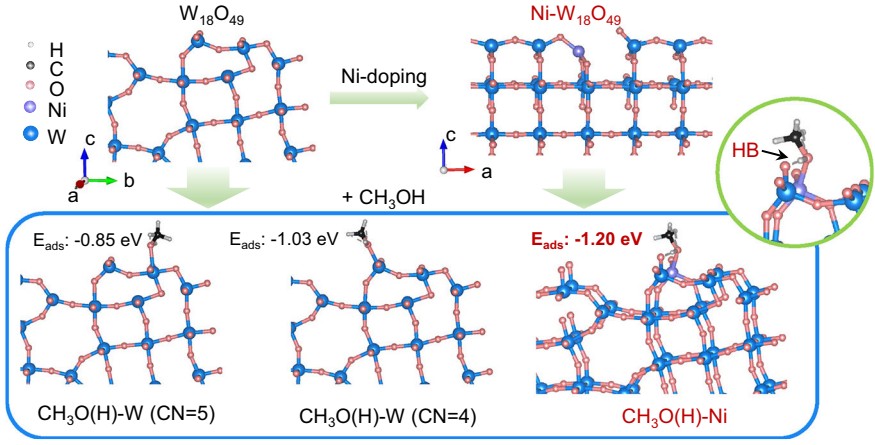

**Fig. 1 | Simulated methanol adsorption on catalysts.** Density functional theory (DFT) calculations were used to simulate the adsorption energies of methanol on low-coordinated W sites (CN = 4 and 5) of $W_{18}O_{49}$, as well as on the Ni site of Ni-doped $W_{18}O_{49}$. *HB* hydrogen bond, *CN* coordination number.

methanol activation pathways. Low-coordinated W sites with coordination numbers (CN) of 4 and 5 were identified as primary methanol adsorption sites, with energies of −1.03 eV and −0.85 eV, respectively. These sites induce significant C-O bond elongation from 1.432 Å to 1.459 Å (CN = 5) and 1.464 Å (CN = 4), confirming their roles in bond activation. Upon introducing various metal atoms (Au, Ag, Cu, Pd, Pt, and Ni) into $W_{18}O_{49}$ for simulations (Supplementary Fig. 1), Ni-doping was found to disrupt the surface structure of $W_{18}O_{49}$, generating W = O dangling bonds and Ni site (Fig. 1). DFT simulations reveal that the Ni site and the adjacent W = O can synergistically adsorb methanol through Ni-O bonding and hydrogen bond (HB) interaction, with an energy of −1.20 eV. The O-H bond (0.977 Å) of methanol elongates to 1.015 Å, suggesting that the Ni site can activate the O-H bond for cleavage. This finding demonstrates the potential of doped Ni as active sites for the C-O-C coupling reaction. Therefore, constructing W and Ni dual active sites on plasmonic $W_{18}O_{49}$ without compromising its surface activity and SPR intensity is a promising strategy yet a significant challenge to enhance photocatalytic methanol dehydration for DME generation.

Herein, we engineered W and Ni dual active sites on plasmonic $W_{18}O_{49}$ through one-step Ni doping, which significantly enhances photocatalytic methanol dehydration for DME production. The localized plasmon on the surface of the catalyst generates hot electrons, which amplify the synergistic effect of W and Ni sites to facilitate C-O bond cleavage and C-O-C coupling, collectively driving the methanol-to-DME conversion pathway. As a result, the reaction activation energy ($E_a$) over the optimized $Ni_{0.66}$-$W_{18}O_{49}$ catalyst is reduced to 65.1 kJ mol$^{-1}$ under 400 mW cm$^{-2}$ irradiation, much lower than 142.4 kJ mol$^{-1}$ of thermocatalysis. This significant reduction in $E_a$ improves the DME generation rate to 133.7 ± 3.3 mmol g$^{-1}$ h$^{-1}$. The $E_a$ can be further reduced to 23.4 kJ mol$^{-1}$ as the irradiation intensity increases to 1000 mW cm$^{-2}$, enhancing the DME generation rate to 784.7 mmol g$^{-1}$ h$^{-1}$. The dual-active-site architecture demonstrates broad applicability, enhancing intramolecular alcohol dehydration for alkene generation by 40–80% and achieving a recorded isobutylene production rate of 3.7 mol g$^{-1}$ h$^{-1}$. Our work demonstrates the superiority of plasmonic semiconductor systems in boosting chemical transformation.

## Results and discussion
### Photocatalytic methanol dehydration over plasmonic $Ni_x$-$W_{18}O_{49}$

To effectively incorporate Ni atoms into plasmonic $W_{18}O_{49}$, Ni-doped $W_{18}O_{49}$ was synthesized via a one-step solvothermal process, utilizing $W(CO)_6$, $NiCl_2$, and hydrochloric acid as precursors in ethanol

solution[20]. Various amounts of Ni were incorporated into $W_{18}O_{49}$, designated as $Ni_x$-$W_{18}O_{49}$ (where x represents the actual weight percentage of doped Ni measured by inductively coupled plasma mass spectrometry (ICP-MS), Supplementary Table 1), to investigate the relationship between doped Ni and W atoms. For comparative studies, control samples including Co-, Pd-, Pt-, Au-, Ag-, and Cu-$W_{18}O_{49}$ with varying dopant concentrations were also synthesized.

The photocatalytic performance of $Ni_x$-$W_{18}O_{49}$ was evaluated by the methanol dehydration reaction (Fig. 2a), which is a primary catalytic process for DME generation. As depicted in Fig. 2b, the optimized $Ni_{0.66}$-$W_{18}O_{49}$ photocatalyst exhibits a notably high production rate of DME (133.7 ± 3.3 mmol g$^{-1}$ h$^{-1}$) compared to the $W_{18}O_{49}$ (13.2 ± 1.2 mmol g$^{-1}$ h$^{-1}$), and the maximum methanol conversion rate reaches 41.5% per hour (Supplementary Fig. 2). DME is the main product with a selectivity of 97.6-98.8%, and small amounts of $H_2$, $CH_4$, and CO were detected by gas chromatography-mass spectrometry (GC-MS) during photocatalytic methanol dehydration over $Ni_x$-$W_{18}O_{49}$ (Supplementary Fig. 3). The generation of $H_2O$ during photocatalysis was confirmed by the $^1H$ nuclear magnetic resonance spectroscopy (NMR, Supplementary Fig. 4). The molar ratio between consumed methanol and generated DME is 2.03 (Supplementary Fig. 5), proving the carbon balance during the photocatalytic methanol dehydration reaction. The photocatalytic methanol dehydration stability of $Ni_{0.66}$-$W_{18}O_{49}$ was evaluated by 21 consecutive cycles (Fig. 2c), with results indicating negligible decay in DME yield. The comparisons of activity and selectivity for methanol dehydration over a range of metal-doped $W_{18}O_{49}$ catalysts are illustrated in Fig. 2d and Supplementary Fig. 6. The optimal DME production rates of Co-, Pd-, Pt-, Au-, Ag-, and Cu-$W_{18}O_{49}$ are 19.6, 30.6, 126.4, 95.7, 67.7, and 43.6 mmol g$^{-1}$ h$^{-1}$, respectively, which are all lower than that of $Ni_{0.66}$-$W_{18}O_{49}$. The actual contents of doped metals in the optimal Co-, Pd-, Pt-, Au-, Ag-, and Cu-$W_{18}O_{49}$ samples were measured by ICP-MS (Supplementary Table 2), which are 0.61 wt%, 0.73 wt%, 1.04 wt%, 0.96 wt%, 0.75 wt%, and 0.69 wt%, respectively. The disparity underscores the distinctive catalytic properties of $W_{18}O_{49}$ conferred by Ni doping. Furthermore, the $Ni_{0.66}$-$W_{18}O_{49}$ catalyst displays high activity in photocatalytic dehydration of $C_{2+}$ alcohols to alkenes (Supplementary Fig. 7). As shown in Figs. 2e, f, compared to $W_{18}O_{49}$, the photocatalytic alkene production rates of $Ni_{0.66}$-$W_{18}O_{49}$ are significantly enhanced, with the yields of ethylene from ethanol dehydration, propylene from propanol dehydration, propylene from isopropanol dehydration, butylene from 1-butanol dehydration, isobutylene from isobutanol dehydration, butylene and 2-butylene from 2-butanol dehydration, and isobutylene from tert-butanol dehydration increasing by 82%

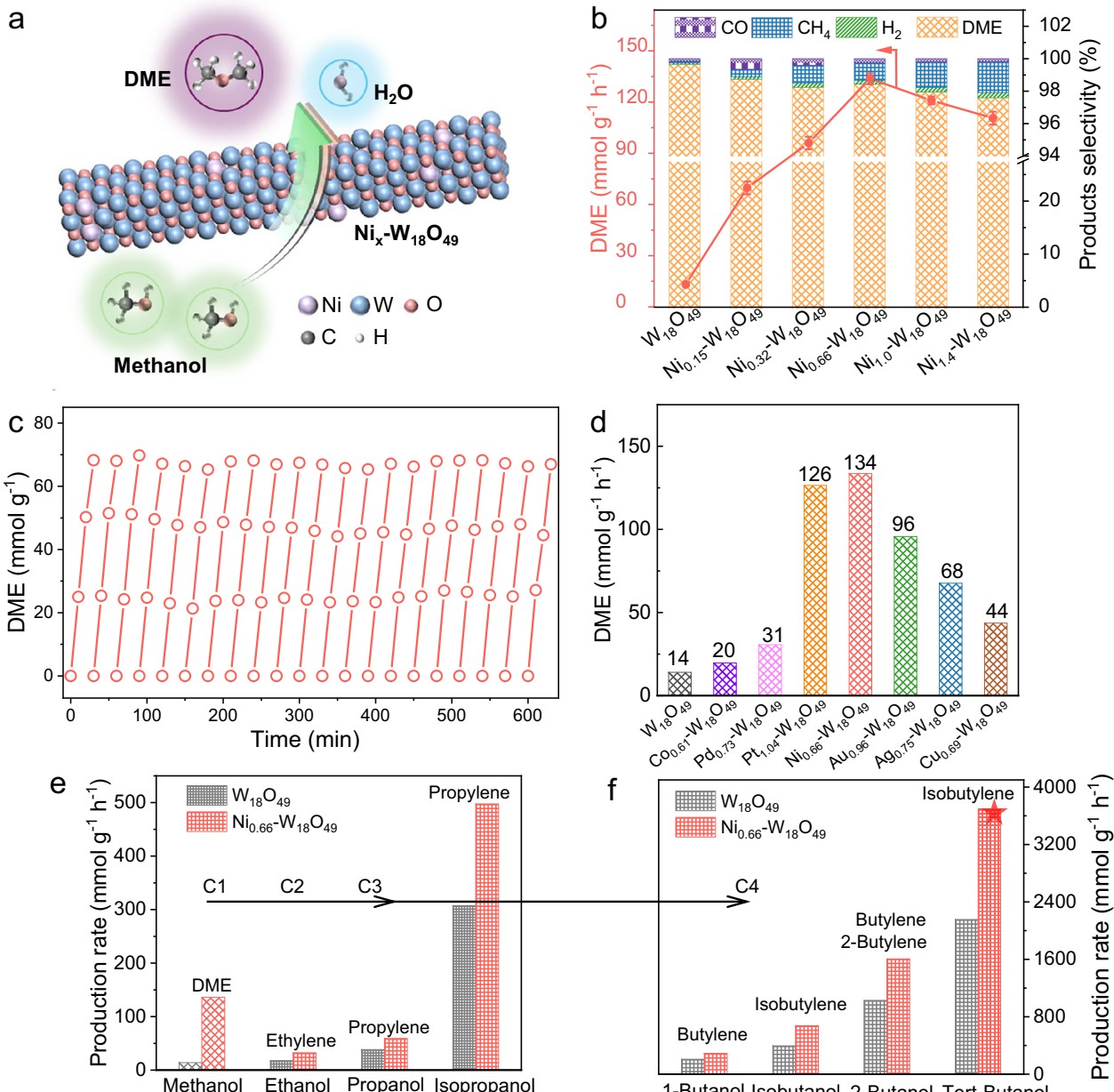

**Fig. 2 | Photocatalytic methanol dehydration. a** Schematic diagram of photocatalytic methanol dehydration. **b** DME generation rate and product selectivity over $Ni_x$-$W_{18}O_{49}$. Error bars correspond to the standard deviation determined from three independent measurements. **c** 21-times recycled photocatalytic methanol dehydration reaction over $Ni_{0.66}$-$W_{18}O_{49}$. **d** DME generation rates over different element-doped $W_{18}O_{49}$ (Pd-$W_{18}O_{49}$, Pt-$W_{18}O_{49}$, Co-$W_{18}O_{49}$, Ni-$W_{18}O_{49}$, Au-$W_{18}O_{49}$, Ag-$W_{18}O_{49}$, and Cu-$W_{18}O_{49}$). **e, f** Photocatalytic dehydration reactions of other higher alcohols (C2, C3, and C4) over $Ni_{0.66}$-$W_{18}O_{49}$ for alkene generation. Light source: 400 mW cm$^{-2}$ supplied by a 300 W Xe lamp (Perfectlight, PLS-SXE300D) equipped with an AM 1.5 G filter. The measurements in Figs. 2d, e, f were only performed once.

(32.4 mmol g$^{-1}$ h$^{-1}$), 54% (59.1 mmol g$^{-1}$ h$^{-1}$), 62% (497.6 mmol g$^{-1}$ h$^{-1}$), 40% (290.3 mmol g$^{-1}$ h$^{-1}$), 71% (675.4 mmol g$^{-1}$ h$^{-1}$), 56% (1607.5 mmol g$^{-1}$ h$^{-1}$), and 71% (3694.6 mmol g$^{-1}$ h$^{-1}$), respectively. Notably, a record isobutylene production rate of 3.7 mol g$^{-1}$ h$^{-1}$ was achieved in photocatalytic tert-butanol dehydration, highlighting the pivotal role of Ni-$W_{18}O_{49}$ in enhancing alcohol dehydration reactions.

The influence of light intensity on the photocatalytic methanol dehydration reaction has been meticulously investigated. As shown in Fig. 3a, a direct correlation between DME production rates and light intensity is observed, with a significant increase from 25.9 mmol g$^{-1}$ h$^{-1}$ to 784.7 mmol g$^{-1}$ h$^{-1}$ as the light intensity increased

from 200 to 1000 mW cm$^{-2}$. Concurrently, the methanol conversion improves dramatically from 10.4% to 94.9% per hour. Supplementary Fig. 8 displays that the photothermal effect of $Ni_{0.66}$-$W_{18}O_{49}$ causes a substantial increase in the surface temperature from 92 to 175 °C. In contrast, the DME production rates of the thermocatalytic reaction are 0.027 to 1.87 mmol g$^{-1}$ h$^{-1}$ at the temperature from 92 to 175 °C, which are far lower than those of the photocatalytic reaction. Compared to the thermocatalytic performance of various reported catalysts shown in Supplementary Table 3, the photocatalytic performance of $Ni_{0.66}$-$W_{18}O_{49}$ is highly competitive in DME generation.

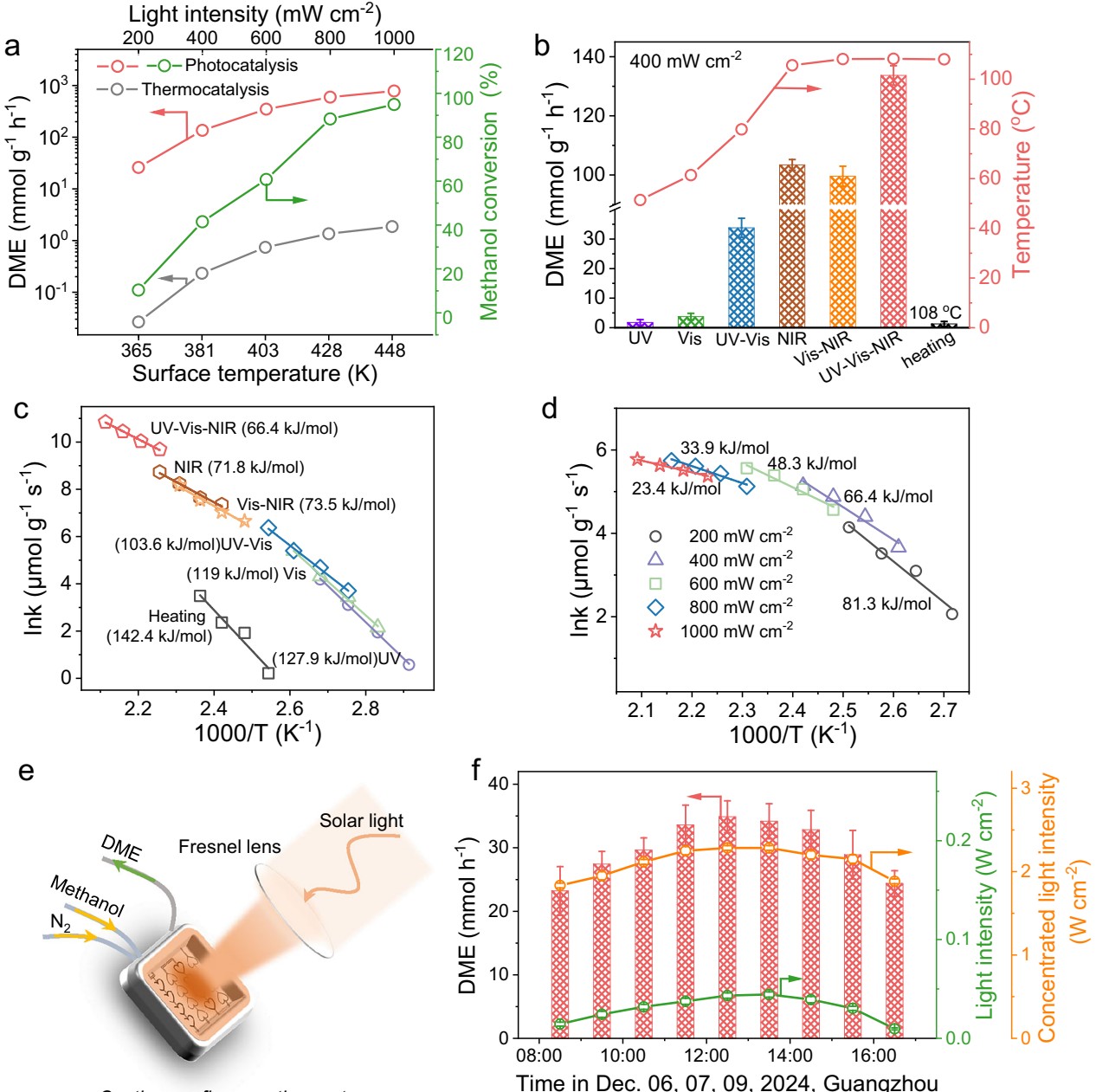

**Fig. 3 | Effect of plasmonic photothermal and hot electrons on the dehydration reaction and $E_a$. a** Photocatalytic DME generation rates and methanol conversion over $Ni_{0.66}$-$W_{18}O_{49}$ under different light intensities, and thermocatalytic DME generation rates at the corresponding temperatures. **b** Surface temperatures of $Ni_{0.66}$-$W_{18}O_{49}$ and its photocatalytic DME generation rates under different light irradiations (UV, Vis, UV-Vis, NIR, Vis-NIR, and UV-Vis-NIR, with a fixed intensity of 400 mW cm$^{-2}$). Error bars correspond to the standard deviation determined from three independent measurements. **c, d** Arrhenius plots and the calculated $E_a$ for methanol dehydration under different light irradiations (**c**) and AM 1.5 G light irradiation with different densities (200, 400, 600, 800, and 1000 mW cm$^{-2}$) (**d**). **e, f** Continuous-flow reaction system (**e**) and DME production rates (**f**) over $Ni_{0.66}$-$W_{18}O_{49}$ under concentrated natural solar light irradiation (the photocatalyst loading in the continuous-flow reaction system is 50 mg). Error bars correspond to the standard deviation determined from three independent measurements.

Investigations into the photothermal effects of $Ni_{0.66}$-$W_{18}O_{49}$ induced by various light sources (with a fixed intensity of 400 mW cm$^{-2}$) are displayed in Fig. 3b and Supplementary Fig. 9. The surface temperatures of $Ni_{0.66}$-$W_{18}O_{49}$ under UV, visible (Vis), and combined UV-Vis irradiation were measured to be 51, 61, and 80 °C, respectively. Notably, the surface temperatures of $Ni_{0.66}$-$W_{18}O_{49}$ rose to 106 °C under near-infrared (NIR), and 108 °C under Vis-NIR or full-spectrum (UV-Vis-NIR) irradiation. The significantly increased surface temperature demonstrates the predominant role of SPR in the NIR region in driving the photothermal effect. Moreover, the DME generation rates of $Ni_{0.66}$-$W_{18}O_{49}$ under different light irradiations are

measured. Figure 3b shows that $Ni_{0.66}$-$W_{18}O_{49}$ exhibits a low DME generation rate under UV (1.8 ± 0.8 mmol g$^{-1}$ h$^{-1}$) or Vis (3.7 ± 1.1 mmol g$^{-1}$ h$^{-1}$) irradiation. However, the synergistic effect of UV and Vis light irradiation effectively enhances the DME generation rate to 33.8 ± 3.2 mmol g$^{-1}$ h$^{-1}$. The activity enhancement is further anabatic under NIR, Vis-NIR, and UV-Vis-NIR irradiation, with DME generation rates of 103.4 ± 1.8, 99.6 ± 3.4, and 133.7 ± 3.3 mmol g$^{-1}$ h$^{-1}$, respectively. These findings strongly suggest that the strong SPR of $Ni_{0.66}$-$W_{18}O_{49}$ in the NIR region plays a vital role in photocatalytic methanol dehydration for DME generation. For reference, the thermocatalytic reaction at 108 °C only yields a mere DME (1.3 ± 0.8 mmol g$^{-1}$ h$^{-1}$), which shows a

stark contrast to the photocatalytic performance. These results indicate the dominant contribution of plasmonic hot carriers in the superior photocatalytic methanol dehydration activity of $Ni_{0.66}$-$W_{18}O_{49}$.

The $E_a$ of methanol dehydration for DME generation over $Ni_{0.66}$-$W_{18}O_{49}$ under various temperatures and different light irradiation conditions (with a fixed intensity of $400\,mW\,cm^{-2}$) is further explored by the Arrhenius equation $k = Ae^{-E_a/RT}$. Figure 3c illustrates the DME generation rates measured under different conditions, and the resulting Arrhenius plots reveal substantial differences in $E_a$. Under pure thermal condition, the $E_a$ of methanol dehydration over $Ni_{0.66}$-$W_{18}O_{49}$ is determined to be $142.4\,kJ\,mol^{-1}$. Under UV, Vis, and UV-Vis light irradiation, the $E_a$ is reduced to 127.9, 119, and $103\,kJ\,mol^{-1}$, respectively. Notably, NIR and Vis-NIR light irradiations further lower the $E_a$ to 71.8 and $73.5\,kJ\,mol^{-1}$, respectively, indicating the significant role of plasmonic hot electrons in reducing the $E_a$ of methanol dehydration. Full-spectrum light irradiation enhances hot electron generation, leading to a greatly reduced $E_a$ of $66.4\,kJ\,mol^{-1}$, which is 52.3% lower than that of thermocatalysis. Furthermore, the influence of light intensity on $E_a$ was explored by measuring the methanol dehydration reaction kinetics. As shown in Fig. 3d and Supplementary Fig. 10, the calculated $E_a$ decreases linearly with increasing AM 1.5 G light intensity. This trend suggests that higher light intensity generates more plasmonic hot electrons, which in turn significantly lower the $E_a$ of methanol dehydration. The lowest $E_a$ of $23.4\,kJ\,mol^{-1}$ is achieved under $1000\,mW\,cm^{-2}$ irradiation. As an exothermic reaction ($\Delta H$, $-23.9\,kJ\,mol^{-1}$), the light energy is mainly used to reduce the $E_a$ of methanol dehydration and promote the kinetics of DME generation. Photocatalytic and thermocatalytic dehydration $E_a$ of different alcohols were measured, as shown in Supplementary Fig. 11, further confirming the reduced $E_a$ under light irradiation. Consequently, these findings confirm the dominant contribution of plasmonic hot electrons to the photocatalytic methanol dehydration performance of $Ni_x$-$W_{18}O_{49}$.

We designed and constructed a continuous-flow reaction system to assess the photocatalytic methanol dehydration performance of $Ni_x$-$W_{18}O_{49}$ under solar light irradiation (Fig. 3e), and the experimental setup was given in Supplementary Fig. 12. The $Ni_{0.66}$-$W_{18}O_{49}$ catalyst was integrated into a microfluidic reactor with a total volume of 2.5 mL. Methanol was injected into the reactor with nitrogen gas serving as the carrier, with an optimized flow rate of $22.4\,mL\,min^{-1}$. Solar light, concentrated by a Fresnel lens, was directly irradiated on the reaction system to facilitate the methanol dehydration reaction, and the generated DME was collected for analysis. The performance of this continuous-flow system was evaluated at Jinan University Panyu Campus, Guangzhou, China, on December 06, 07, and 09, 2024. Figure 3f illustrates that the focused solar light intensity was $1.83 \pm 0.03\,W\,cm^{-2}$ at 8:30 AM, reaching a peak of $2.28 \pm 0.03\,W\,cm^{-2}$ at noon. In response, the DME production rate increased from $23.2 \pm 3.8\,mmol\,h^{-1}$ to a maximum of $34.9 \pm 2.5\,mmol\,h^{-1}$, demonstrating the practical applicability and high efficiency of $Ni_{0.66}$-$W_{18}O_{49}$ under solar light irradiation.

## Structural characterizations of catalysts $Ni_x$-$W_{18}O_{49}$

Pristine $W_{18}O_{49}$ displays a sea urchin-like morphology, with preferential growth along the <010> direction[21], as shown in Fig. 4a and Supplementary Fig. 13a. High-angle annular dark-field scanning transmission electron microscopy (HAADF-STEM) images in Fig. 4b reveal a uniform atomic distribution throughout the $W_{18}O_{49}$ structure. In contrast, the $Ni_x$-$W_{18}O_{49}$ catalysts exhibit no significant morphological changes as the Ni content increases from 0.15 to 1.4 wt% (Supplementary Fig. 13b–f). However, a noticeable disruption in the surface structure is observed for $Ni_{0.66}$-$W_{18}O_{49}$ (Fig. 4c), suggesting that Ni doping induces surface alterations. Despite this, the HAADF-STEM image in Fig. 4d does not clearly resolve the distribution of Ni atoms,

likely due to the lower atomic number of Ni (28) than W (74). Energy-dispersive X-ray spectroscopy (EDS) elemental mapping, as shown in Fig. 4e–h, confirms the homogeneous distribution of Ni atoms in the $W_{18}O_{49}$ nanowires. The crystallinity of the $Ni_x$-$W_{18}O_{49}$ catalysts was investigated using X-ray diffraction (XRD), as shown in Fig. 4i, and a shift in the [010] peak to a lower angle indicates a structural change caused by Ni doping. Raman spectra, as shown in Fig. 4j and Supplementary Fig. 14, further demonstrate the structural change, with shifts of both v(O-W-O) and δ(O-W-O) modes to lower wavenumbers in $Ni_{0.66}$-$W_{18}O_{49}$. Notably, a new band at $963.8\,cm^{-1}$ is observed in $Ni_x$-$W_{18}O_{49}$ catalysts, with the enhanced intensity as Ni doping increases, which is assigned to the surface terminal W=O stretching mode[22]. Low-valent $Ni^{2+}$ substitutes for high-valent $W^{6+}$, $W^{5+}$, or $W^{4+}$, and oxygen vacancies will spontaneously form to maintain charge balance in $Ni_x$-$W_{18}O_{49}$. This indicates that the surface-ordered crystal structure of $W_{18}O_{49}$ is disrupted by Ni doping, as shown in 4j, leading to the loss of some lattice oxygen atoms and the creation of coordinatively unsaturated W atoms on the surface[23].

The intrinsic crystal structure of $W_{18}O_{49}$ has an abundance of low-valent W ions ($W^{4+}$ and $W^{5+}$), leading to its high electron density. This characteristic is evident in electron paramagnetic resonance (EPR) spectra, as shown in Supplementary Fig. 15. As the increase of Ni content in $Ni_x$-$W_{18}O_{49}$ from 0.15 to 1.4 wt%, there is a slight decrease in electron density. X-ray photoelectron spectroscopy (XPS) was employed to study the chemical states of W atoms in $Ni_x$-$W_{18}O_{49}$, and the contents of $W^{5+}$ and $W^{4+}$ in $Ni_x$-$W_{18}O_{49}$ are found to be similar to that in pristine $W_{18}O_{49}$, as detailed in Supplementary Figs. 16, 17. The doped Ni in $Ni_x$-$W_{18}O_{49}$ is detected as $Ni^{2+}$ ion in their Ni 2p XPS spectra (Supplementary Fig. 18). Although the SPR intensity of $Ni_x$-$W_{18}O_{49}$ experiences a slight reduction in the UV-Vis-NIR diffuse reflectance spectroscopy (DRS), the overall SPR characteristics remain robust (Fig. 4k, Supplementary Fig. 19). The broad SPR band of $W_{18}O_{49}$ nanowires is due to the coupling of nanowires, demonstrated by finite-difference time-domain (FDTD) simulations (Supplementary Fig. 20). EPR spectra (Fig. 4l and Supplementary Fig. 21) show that $Ni_{0.66}$-$W_{18}O_{49}$ has a significantly increased electron density after light irradiation, suggesting that an appropriate amount of Ni-doping facilitates photoelectron accumulation on $Ni_{0.66}$-$W_{18}O_{49}$[24]. The influence of light irradiation on the chemical states of Ni and W elements in $Ni_x$-$W_{18}O_{49}$ was analyzed by XPS spectra, and the results in Supplementary Fig. 22 show that no significant change is found in the Ni 2p XPS spectra of $Ni_{0.66}$-$W_{18}O_{49}$ after light irradiation, indicating that electron accumulation does not occur on Ni atoms. Notably, the variations in $W^{5+}$ and $W^{4+}$ contents are clearly depicted in Fig. 4m and Supplementary Fig. 23 (the W 4 f XPS spectra). The $W^{5+}$ content in $W_{18}O_{49}$, $Ni_{0.66}$-$W_{18}O_{49}$, and $Ni_{1.4}$-$W_{18}O_{49}$ similarly increases from 8.0% to 15.1% after light irradiation, while $Ni_{0.66}$-$W_{18}O_{49}$ exhibits the highest $W^{4+}$ content of 14.1%. This result demonstrates that doping of 0.66 wt% Ni in $W_{18}O_{49}$ facilitates photoelectron trapping by high-valent $W^{6+}$ or $W^{5+}$ and generates more low-valent $W^{4+}$ sites, thereby enhancing electron density for strong SPR. The SPR enhancement under light irradiation is more pronounced for $Ni_{0.66}$-$W_{18}O_{49}$ compared to pristine $W_{18}O_{49}$, as evidenced by their enhanced UV-Vis-NIR DRS, as shown in Supplementary Fig. 24. The surface low-valent W atoms can strongly adsorb alcohols, which is crucial for photocatalytic alcohol dehydration[25]. Additionally, the stability of the structure and SPR of $Ni_{0.66}$-$W_{18}O_{49}$ after photocatalysis was confirmed by XRD patterns (Supplementary Fig. 25) and UV-Vis-NIR DRS analysis (Supplementary Fig. 26).

To further elucidate the electronic state and the coordination environment of Ni and W atoms in $Ni_x$-$W_{18}O_{49}$, we have employed X-ray absorption near-edge structure (XANES) and extended X-ray absorption fine structure (EXAFS) spectroscopies. Figure 5a displays the normalized Ni K-edge XANES spectra for $Ni_{0.66}$-$W_{18}O_{49}$ and $Ni_{1.4}$-$W_{18}O_{49}$, with NiO and Ni foil as references. The energy absorption edges of doped Ni in $Ni_{0.66}$-$W_{18}O_{49}$ and $Ni_{1.4}$-$W_{18}O_{49}$ closely resemble

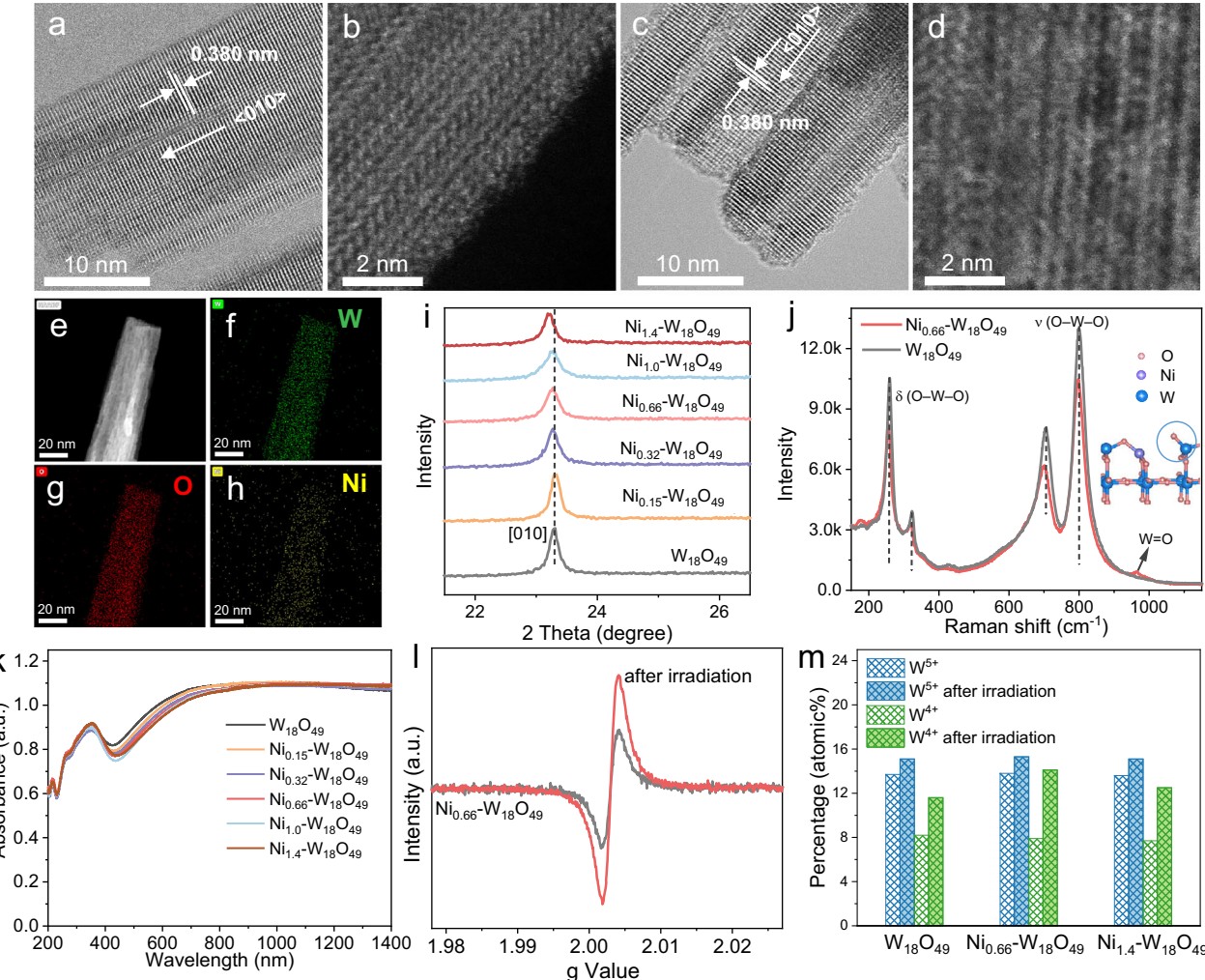

**Fig. 4 | Characterization of catalysts with various analysis techniques. a–d** High resolution transmission electron microscope (HRTEM) and HAAD-STEM images of $W_{18}O_{49}$ (a, b) and $Ni_{0.66}$-$W_{18}O_{49}$ (c, d). **e** Dark-field STEM and **f–h** EDS element (W, O, and Ni) mapping images of $Ni_{0.66}$-$W_{18}O_{49}$. **i** XRD pattern, **j** Raman spectra (insert figure is the surface structure model), and **k** UV-Vis-NIR DRS of $W_{18}O_{49}$ and $Ni_x$-$W_{18}O_{49}$. **l** EPR spectra of $Ni_{0.66}$-$W_{18}O_{49}$ before and after light irradiation. **m** Ratio varies of low-valent W ($W^{5+}$, $W^{4+}$) in $W_{18}O_{49}$, $Ni_{0.66}$-$W_{18}O_{49}$, and $Ni_{1.4}$-$W_{18}O_{49}$ before and after light irradiation.

that of NiO, indicating its oxidation state is close to $Ni^{2+}$[20,26]. As depicted in Fig. 5b, the Fourier-transformed EXAFS (FT-EXAFS) spectra reveal distinct features for the references and the Ni-$W_{18}O_{49}$ samples. The Ni foil exhibits a predominant peak at 2.16 Å, corresponding to the Ni-Ni bond. The NiO reference displays two peaks, a strong peak at 2.54 Å is attributed to the Ni-Ni bond, and the weak peak at 1.64 Å is assigned to the Ni-O bond[27,28]. Comparatively, $Ni_{0.66}$-$W_{18}O_{49}$ only shows a strong peak at 1.52 Å, attributed to the Ni-O bond, which suggests that Ni is atomically dispersed within $Ni_{0.66}$-$W_{18}O_{49}$. However, $Ni_{1.4}$-$W_{18}O_{49}$ exhibits two peaks of similar intensity at 1.52 and 2.62 Å, corresponding to the Ni-O bond of single Ni atom doping and the Ni-Ni bond characteristic of NiO[28,29], respectively, which implies that Ni exists in two distinct coordination environments in $Ni_{1.4}$-$W_{18}O_{49}$. Quantitative structural parameters derived from the Ni K-edge EXAFS fitting curves of $Ni_{0.66}$-$W_{18}O_{49}$, $Ni_{1.4}$-$W_{18}O_{49}$, NiO, and Ni foils are illustrated in Fig. 5c, d, and Supplementary Fig. 27, and the detailed results are tabulated in Supplementary Table 4. The CN of the Ni-O bond in $Ni_{0.66}$-$W_{18}O_{49}$ is determined to be 6.37. For $Ni_{1.4}$-$W_{18}O_{49}$, the CN of Ni-O and Ni-Ni bonds is 6.3 and 6.2, respectively. The reduced CN of the Ni-Ni bond in $Ni_{1.4}$-$W_{18}O_{49}$, compared to the CN (12) of the Ni-Ni bond in NiO, suggests that Ni atoms in $Ni_{1.4}$-$W_{18}O_{49}$ are closely spaced. Wavelet transforms (WT) of the EXAFS spectra, shown in Fig. 5e–h, were conducted to further elucidate the localized structure in both the K and R

spaces. The WT intensity maximum for $Ni_{0.66}$-$W_{18}O_{49}$ at 5.8 Å$^{-1}$ in the K space and 1.52 Å in the R space corresponds to the Ni-O bond, which is distinct from the Ni-Ni bond observed in NiO and Ni foil references. For $Ni_{1.4}$-$W_{18}O_{49}$, an additional WT intensity maximum is observed at 6.8 Å$^{-1}$ in the K space and 2.62 Å in the R space, corresponding to the Ni-Ni bond.

The normalized W L$_3$-edge XANES spectra for $W_{18}O_{49}$ and $Ni_{0.66}$-$W_{18}O_{49}$, with $WO_3$ and W foil as references[30], are shown in Fig. 5i. The white lines of W in $W_{18}O_{49}$ and $Ni_{0.66}$-$W_{18}O_{49}$ nearly overlap, indicating that Ni doping does not alter the valence of W in $W_{18}O_{49}$. The FT-EXAFS spectra, as depicted in Fig. 5j, reveal a strong peak at 1.28 Å and a weak peak at 1.84 Å assigned to two different W-O bonds (W-O1 and W-O2) in the $W_{18}O_{49}$ crystal[18,31]. Quantitative structural parameters derived from the W L$_3$-edge EXAFS fitting curves for $W_{18}O_{49}$ and $Ni_{0.66}$-$W_{18}O_{49}$ are illustrated in Fig. 5k and Supplementary Fig. 28, and the detailed results are tabulated in Supplementary Table 5. The CN of W-O2 in $Ni_{0.66}$-$W_{18}O_{49}$ is slightly lower than that of W-O2 in $W_{18}O_{49}$, and a lower CN of W is favorable to photocatalysis[23,32].

### Photocatalytic methanol dehydration mechanism study on $Ni_x$-$W_{18}O_{49}$

The photocatalytic mechanism of methanol dehydration on $Ni_x$-$W_{18}O_{49}$ has been elucidated by in-situ XPS measurement, as shown in

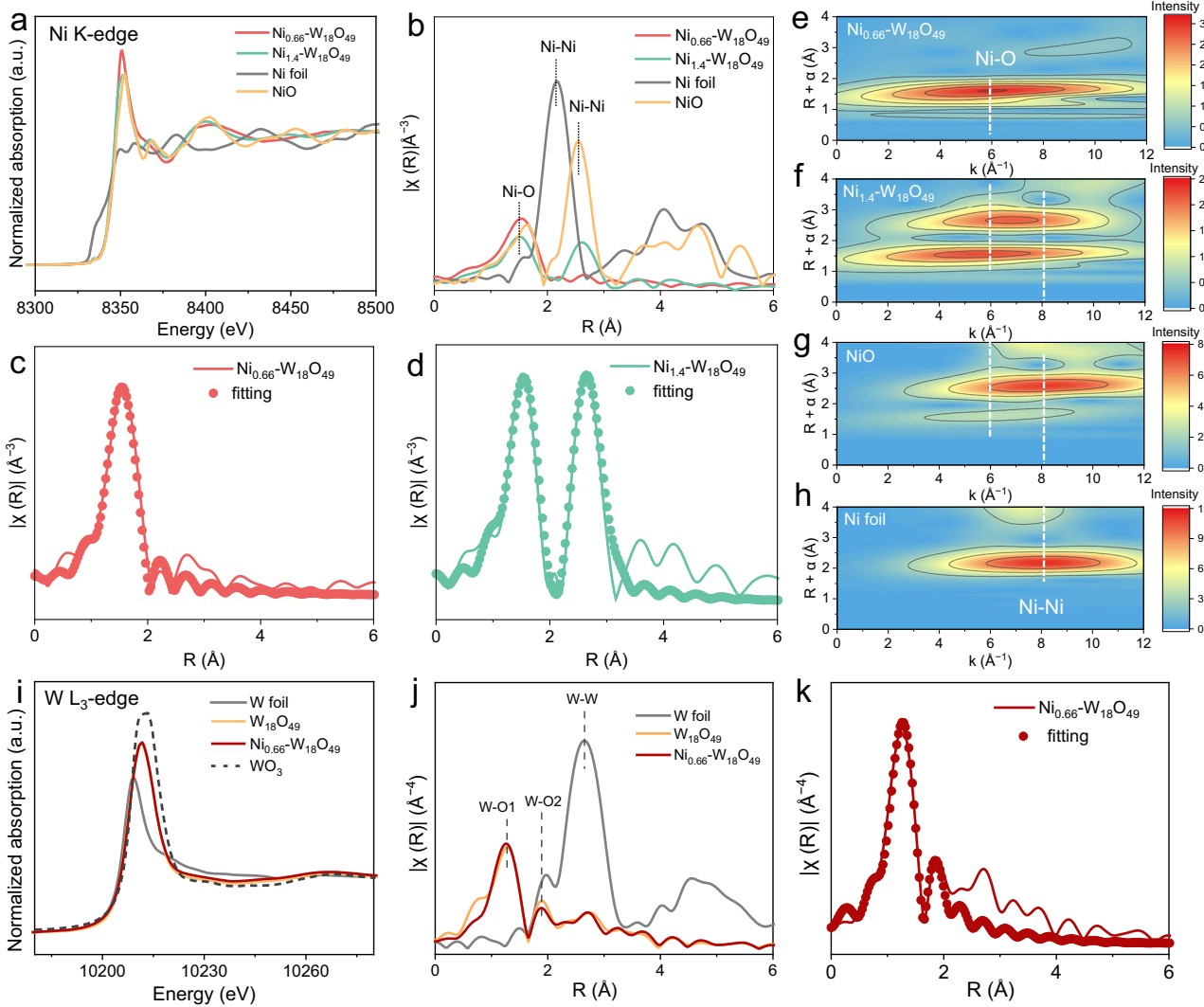

**Fig. 5 | The coordination environment of Ni and W atoms analyzed by XANES.** **a** Normalized Ni K-edge XANES and **b** FT-EXAFS spectra of $Ni_{0.66}$-$W_{18}O_{49}$, $Ni_{1.4}$-$W_{18}O_{49}$, NiO, and Ni foil references. **c**, **d** The Ni K-edge EXAFS fitting curves for $Ni_{0.66}$-$W_{18}O_{49}$ (**c**) and $Ni_{1.4}$-$W_{18}O_{49}$ (**d**). **e**–**h** WT-EXAFS of Ni K-edge for $Ni_{0.66}$- $W_{18}O_{49}$, $Ni_{1.4}$-$W_{18}O_{49}$, NiO, and Ni foil references. **i** Normalized W $L_3$-edge XANES and **j** FT-EXAFS spectra of $W_{18}O_{49}$, $Ni_{0.66}$-$W_{18}O_{49}$, $WO_3$, and Ni foil references. **k** The W $L_3$-edge EXAFS fitting curves for $Ni_{0.66}$-$W_{18}O_{49}$.

Fig. 6a. After the addition of methanol, a distinct peak at 286.5 eV, corresponding to the C-O bond, emerges in the C 1 s XPS spectrum of $Ni_{0.66}$-$W_{18}O_{49}$[33]. The C-O bond peak is more pronounced than that of pristine $W_{18}O_{49}$ (Supplementary Fig. 29), highlighting the enhanced methanol adsorption capacity of $Ni_{0.66}$-$W_{18}O_{49}$. Upon light irradiation, the C-O bond peak diminishes, indicating the C-O bond cleavage of adsorbed methanol for the dehydration reaction, and this process takes place more slowly on $W_{18}O_{49}$ (Supplementary Fig. 29). Further insights come from the in-situ W 4 f XPS spectra, where the adsorbed methanol induces a shift in the W 4 f XPS peak to lower binding energy due to the electron-donating nature of methanol. The shift is more pronounced in $Ni_{0.66}$-$W_{18}O_{49}$, confirming its superior methanol adsorption. Concurrently, the $W^{5+}$ content decreases, suggesting that methanol adsorption occurs on surface low-coordinated W sites. Post-irradiation, the $W^{5+}$ content returns to its initial level within 10 min, while the $W^{4+}$ content increases from 9.1% to 14.7%, and further to 18.7% after 20 min of irradiation. The increase in $W^{4+}$ content is attributed to photoelectron accumulation on $Ni_{0.66}$-$W_{18}O_{49}$ in the methanol atmosphere. As light irradiation prolongs, the W 4 f XPS peak shifts back to high binding energy, revealing the consumption of surface-adsorbed methanol for DME generation (Fig. 6a and Supplementary Fig. 30). The

adsorption and dehydration of methanol are also evident by the in-situ O 1 s XPS spectra (Supplementary Fig. 31).

The photocatalytic methanol dehydration processes over $W_{18}O_{49}$ and $Ni_{0.66}$-$W_{18}O_{49}$ were further monitored by in-situ Fourier transform infrared (FT-IR) spectroscopy, as depicted in Fig. 6b. After the addition of methanol, new strong bands around 1010 ~ 1060, 1362 ~ 1462, and 2824 ~ 2984 $cm^{-1}$, assigned to vC-O, δCH₃, and vCH₃[34,35], respectively, demonstrate the strong methanol adsorption on $Ni_{0.66}$-$W_{18}O_{49}$. Under full-spectrum light irradiation, a new band around 1161.9 $cm^{-1}$, corresponding to vC-O-C[36,37], emerges and strengthens with prolonged light irradiation, verifying the methanol dehydration to the DME process. Furthermore, the vC-O band intensities for both $Ni_{0.66}$-$W_{18}O_{49}$ and $W_{18}O_{49}$ increase with prolonged UV-Vis-NIR irradiation, which is attributed to the light-induced more low-valent $W^{4+}$ and $W^{5+}$ (Fig. 4m) for enhanced methanol adsorption. The signal of *OH intermediate was enhanced under full-spectrum light irradiation, and then became weak due to its removal from the catalyst surface as $H_2O$. However, the vC-O-C band is weaker in the in-situ FT-IR spectra of $W_{18}O_{49}$ (Fig. 6c), confirming a higher activity of $Ni_{0.66}$-$W_{18}O_{49}$ for photocatalytic methanol dehydration. When UV light is irradiated on $Ni_{0.66}$-$W_{18}O_{49}$, no vC-O-C band is observed, but as the light switches to NIR, the vC-O-

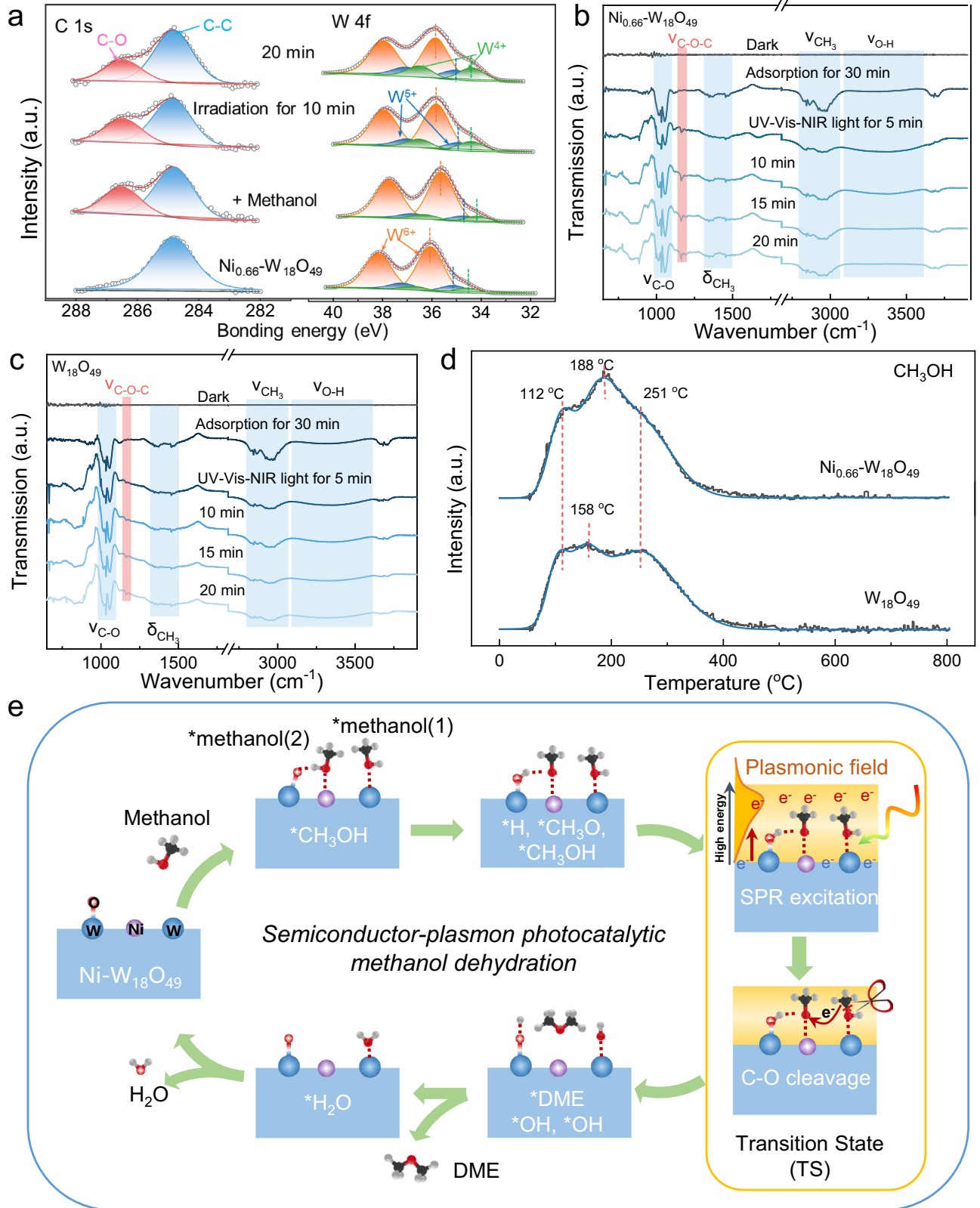

**Fig. 6 | In situ spectroscopy analysis and photocatalytic methanol dehydration mechanism. a** In-situ C 1 s and W 4 f XPS spectra of $Ni_{0.66}$-$W_{18}O_{49}$ during photocatalytic methanol dehydration reaction. **b**, **c** In-situ FT-IR spectra of $Ni_{0.66}$-$W_{18}O_{49}$ and $W_{18}O_{49}$ during photocatalytic methanol dehydration reaction under full-spectrum light irradiation. **d** Methanol-TPD pattern of the $W_{18}O_{49}$ and $Ni_{0.66}$-$W_{18}O_{49}$ catalysts. **e** Photocatalytic mechanism of methanol dehydration reaction over $Ni_x$-$W_{18}O_{49}$.

C band appears and strengthens with prolonged irradiation time (Supplementary Fig. 32), indicating that NIR-excited plasmonic hot electrons boost DME generation. These results demonstrate that the Ni doping in $W_{18}O_{49}$ enhances methanol adsorption, thereby boosting the plasmonic dehydration reaction for DME generation. The reaction pathway of $C_{2+}$ alcohol (tert-butanol) dehydration over $W_{18}O_{49}$ and $Ni_{0.66}$-$W_{18}O_{49}$ was monitored by the in-situ FT-IR spectroscopy, as shown in Supplementary Fig. 33. UV-irradiation can promote more tert-butanol adsorption on $Ni_{0.66}$-$W_{18}O_{49}$, and NIR-irradiation promotes C-O bond cleavage for C = C bond generation. Temperature-programmed desorption (TPD) of methanol on $W_{18}O_{49}$ and $Ni_{0.66}$-$W_{18}O_{49}$ are presented in Fig. 6d. The desorption peak observed at 112 °C and 188 °C corresponds to the physical adsorption and chemical adsorption of methanol, respectively. Notably, in comparison to $W_{18}O_{49}$, the desorption peak at 188 °C on the $Ni_{0.66}$-$W_{18}O_{49}$ sample is significantly enhanced, indicating that the incorporation of Ni sites effectively improves the chemical adsorption capacity for methanol molecules[38]. While excessive Ni doping reduces the adsorption amount of methanol, restraining the dehydration reaction for DME generation (Supplementary Fig. 34).

Based on the results and discussions, a plausible mechanism for photocatalytic methanol dehydration over Ni-$W_{18}O_{49}$ is proposed, as illustrated in Fig. 6e. Ni doping in $W_{18}O_{49}$ induces surface reconstruction, creating W = O dangling bonds. DFT calculations (Fig. 1) have demonstrated that two distinct surface sites existed on Ni-$W_{18}O_{49}$ for methanol adsorption: low-coordinated W and Ni sites. The surface low-coordinated W atom can directly bond to the oxygen of methanol, forming *methanol(1), while the Ni and adjacent W = O have a synergy on adsorbing methanol by Ni-O bonding and a hydrogen bond interaction, resulting in *methanol(2). According to DFT simulations (Supplementary Figs. 35, 36), the O-H bond of *methanol(2) is activated for cleavage to form W-O-H and *CH_3O as intermediates[39]. The excitations on $Ni_{0.66}$-$W_{18}O_{49}$ under UV, Vis-NIR, and UV-Vis-NIR light irradiation were illustrated in Supplementary Fig. 37. Most photoelectrons excited by UV irradiation became trapped at low energy states within conduction band, thereby increasing the electron density. Upon Vis-NIR light irradiation, electrons on low-valent W atoms undergo strong oscillations due to SPR, creating an intense plasmonic field on the surface of $W_{18}O_{49}$ (Supplementary Fig. 38)[18]. SPR also excites low-energy electrons to high-energy hot electrons. Both *methanol(1) and *CH_3O are located in the plasmonic field, and the plasmonic hot electrons can be transferred to *methanol(1) through the W-O bond, activating and cleaving the C-O bond in methanol for *CH_3 generation and leaving OH adsorbed on W sites as W-O-H. However, *CH_3O connected to Ni prevents hot electron transfer and protects its C-O bond. During transition state (TS), the cleavage of the C-O bond is the rate-determining step for the methanol dehydration reaction. The generated *CH_3 from *methanol(1) attacks the oxygen atom of *CH_3O to achieve C-O-C coupling for adsorbed DME (*DME) formation. Ultimately, a dehydration reaction between two W-O-H produces one $H_2O$, and the catalyst returns to its initial state after releasing adsorbed $H_2O$ (*$H_2O$). Meanwhile, a small part of doped Ni can produce a small amount of hydrogen, and the generated *CH_3 can react with *H for $CH_4$ byproduct generation as demonstrated in Fig. 2b. However, the thermal-activation for C-O cleavage and C-O-C coupling is hard due to the large activation energy (142.4 kJ mol⁻¹) for thermocatalytic methanol dehydration. For $W_{18}O_{49}$, most methanol is adsorbed on surface low-coordinated W sites as *methanol(1). During TS, plasmonic hot electron transfer can break the C-O bond of *methanol(1), but the generated *CH_3 is less likely to react with the oxygen atom of another *methanol(1), slowing the DME generation rate. Although DFT calculations still exhibit some discrepancies from actual experiences, they further demonstrate that Ni-doping in $W_{18}O_{49}$ reduces $E_a$ of methanol dehydration as shown in Supplementary Figs. 35, boosting DME generation. Therefore, the synergy of W and Ni sites enhances the plasmonic photocatalytic methanol dehydration to DME. This photocatalytic mechanism underscores the synergistic interaction between distinct surface sites and the amplification effect of the semiconductor-plasmon on $Ni_x$-$W_{18}O_{49}$, thereby effectively facilitating the methanol dehydration process. While for $C_{2+}$ alcohol dehydration, $Ni_{0.66}$-$W_{18}O_{49}$ with more light-induced surface low-valent $W^{4+}$ and $W^{5+}$ can facilitate $C_{2+}$ alcohol adsorption, meanwhile, its hot electrons promote C-O bond cleavage for alkene generation.

In summary, we present a one-step synthesis strategy for engineering dual active sites on plasmonic $W_{18}O_{49}$ through Ni doping, which remarkably enhances the photocatalytic methanol dehydration reaction for DME production. The distinct roles of W and Ni sites in the catalytic process have been elucidated: low-coordinated W sites facilitate C-O bond cleavage promoted by SPR-excited hot electrons and Ni sites enable C-O-C coupling for DME generation. The optimized $Ni_{0.66}$-$W_{18}O_{49}$ catalyst dramatically reduces $E_a$ to 65.1 kJ mol⁻¹ under 400 mW cm⁻² illumination, much lower than 142.4 kJ mol⁻¹ required for thermocatalysis. This significant $E_a$ reduction enables a good DME production rate of 133.7 ± 3.3 mmol g⁻¹ h⁻¹. Further increasing light intensity to 1000 mW cm⁻² lowers $E_a$ to 23.4 kJ mol⁻¹ while boosting the DME generation rate to 784.7 mmol g⁻¹ h⁻¹. This Ni-doped $W_{18}O_{49}$ catalyst also demonstrates broad applicability in various alcohol dehydration for alkene generation, achieving a recorded photocatalytic isobutylene production rate of 3.7 mol g⁻¹ h⁻¹. This work showcases the exceptional potential of rationally designed plasmonic semiconductor systems in driving photochemical transformations, establishing a sustainable and energy-efficient paradigm for DME production.

## Methods

### Materials
Tungsten hexacarbonyl (W(CO)_6, >97%), methanol (99.8%), anhydrous ethanol (99.8%), propanol (99.8%), isopropanol (99.8%), 1-butanol (99.8%), isobutanol (99.8%), 2-butanol (99.8%), tert-butanol (99.8%), NiCl_2.6H_2O (>98%), HAuCl_4.3H_2O (99.9%), H_2PtCl_6.6H_2O (99.9%), CuCl_2.2H_2O (>98%), PdCl_2 (99.9%), CoCl_2.6H_2O (99%), AgNO_3 (99.9%), and hydrochloric acid (HCl, AR, 36%~38%) were purchased from Aladdin and used without further purification. Deionized water (18.2 MΩ) was made by Arium Mini Plus ultrapure water system and used in all experiments.

### Synthesis of catalysts
$Ni_x$-$W_{18}O_{49}$ samples were prepared by using a solvothermal method[20]. In a typical procedure, 150 mg of W(CO)_6 was dissolved in 28.75 mL of ethanol, forming a bright yellow solution. Subsequently, different volumes (0.196, 0.392, 0.588, 0.784, and 0.980 mL) of a NiCl_2·6H_2O solution (10 mg mL⁻¹) and 1.25 mL of concentrated HCl (36%-38%) were added into above solution under vigorous stirring. The resulting mixture was transferred to a 50 mL Teflon-lined stainless-steel autoclave and heated at 180 °C for 12 h. After cooling to room temperature, $Ni_x$-$W_{18}O_{49}$ catalysts were collected after centrifugation, washed thoroughly with ethanol, and dried in a vacuum oven. The other metal-doped $W_{18}O_{49}$ samples, including Au-$W_{18}O_{49}$, Ag-$W_{18}O_{49}$, Cu-$W_{18}O_{49}$, Pt-$W_{18}O_{49}$, Pd-$W_{18}O_{49}$, and Co-$W_{18}O_{49}$ were synthesized following above process using HAuCl_4, AgNO_3, CuCl_2, H_2PtCl_6, PdCl_2, and CoCl_2 precursor solutions, respectively. The pristine $W_{18}O_{49}$ catalyst was prepared as a reference through the above synthetic process without adding the doped metal salts.

### Photocatalytic methanol dehydration reaction tests
A 5 mg catalyst was plastered on a cover glass (9.6 cm²) and placed at the bottom of a 180 mL reaction chamber. The chamber was sealed with a thick quartz cover glass and purged with pure nitrogen for 20 min to remove oxygen. Illumination was supplied by a 300 W Xe lamp (Perfectlight, PLS-SXE300D) equipped with an AM 1.5 G filter, delivering a light intensity of 400 mW cm⁻². The light intensity can be

adjusted by modulating the working current of light source. 0.1 mL of methanol was injected into the chamber for reaction. Gaseous products were analyzed using gas chromatography-mass spectrometry (GC-MS, Panna Instruments A91Plus + AMD9P) equipped with a Plot-Q column. Photocatalytic reaction tests under different light irradiation were measured following the above process, employing different light-cutoff filters for NIR ( > 800 nm), Vis (420–780 nm), UV (200-400 nm), Vis-NIR ( > 420 nm), and UV-Vis (<800 nm) light irradiation. The reaction temperature was controlled by a thermal oil bath.

The flow reaction of photocatalytic methanol dehydration reaction was performed in heart-shaped micropipes engraved on a quartz plate, with a diameter of 1 mm and a total volume of 2.5 mL. 50 mg catalyst was coated on the inner walls of the micropipes, which were sealed with another quartz plate as a reaction chamber. Methanol was injected into the chamber with a flow rate of 50 μL min$^{-1}$ through one inlet, while nitrogen was introduced through a second inlet at a flow rate of 22.4 mL min$^{-1}$ to facilitate methanol flow along the micropipes. Sunlight was focused onto the quartz window of the chamber using a Fresnel lens, covering a total irradiation area of 16 cm$^2$. The gas products were collected for detection by GC-MS.

### The reaction activation energy calculation

The reaction activation energy of methanol dehydration is calculated by the Arrhenius equation as follows[40]:

$$\ln k = \ln A - \frac{E_a}{RT} \tag{1}$$

where $k$ is the reaction rate constant, $A$ is the Arrhenius constant, $E_a$ is the activation energy, $R$ is the molar gas constant, and $T$ is the surface temperature of the catalysts. The surface temperature of the catalysts was detected by a infrared thermal camera (Fluke Tis20 + ).

### DFT calculations

All computational simulations were performed using the DS-PAW software based on density functional theory (DFT)[41]. The interaction between the valence electrons and the ionic cores was described using projector-augmented wave (PAW) pseudopotentials. During the structural optimization, the Perdew-Burke-Ernzerhof (PBE) functional, within the generalized gradient approximation (GGA), was applied. For the $W_{18}O_{49}$ (100) surface model (Supplementary Data 1), a $1 \times 2$ supercell containing 134 atoms was chosen. A vacuum space of approximately 15 Å was introduced above the slab to avoid interactions between periodic images. In simulations involving Ni doping, a single tungsten (W) atom on the surface was replaced by a nickel (Ni) atom within the same supercell (Supplementary Data 1). Only the surface atoms were allowed to relax during the optimization. The computational setup included a cutoff energy (Ecut) of 500 eV, a total energy convergence criterion of $1 \times 10^{-4}$ eV, and a residual stress tolerance of 0.05 eV/Å. Additionally, a $2 \times 1 \times 1$ Monkhorst-Pack k-point mesh was used.

The formation energy of $CH_3OH$ adsorption was calculated using the following expression:

$$Ef = E(CH_3OH - X) - E(X) - E(CH_3OH) \tag{2}$$

where $X$ represents either $W_{18}O_{49}$ (100) or $Ni-W_{18}O_{49}$ (100), and $E(CH_3OH-X)$, $E(CH_3OH)$, and $E(X)$ denote the total energies of $CH_3OH$ adsorbed on X, $CH_3OH$ alone, and pristine X, respectively.

The Gibbs free energy ($\Delta G$) was calculated using the thermodynamic expression:

$$\Delta G = \Delta E - T\Delta S + \Delta E_{ZPE} \tag{3}$$

where $\Delta E_{ZPE}$ denotes the zero-point energy correction and $\Delta S$ represents the entropy change of intermediate species.

Transition states were located using the climbing image nudged elastic band (CI-NEB) method[42]. The calculations employed 8 intermediate images between initial and final states, with a spring constant of 5.0 eV/Å$^2$ and a force convergence threshold of 0.1 eV/Å.

### In-situ FT×IR transmission measurement

In-situ FT-IR measurements were performed using a Thermo Scientific Nicolet iS50 FT-IR spectrometer equipped with a liquid-nitrogen-cooled HgCdTe detector. Spectra were collected with a resolution of 4 cm$^{-1}$ by averaging 32 scans per measurement. The sample was placed in a custom-designed IR reaction chamber suitable for analyzing highly scattered powder samples in diffuse reflection mode. The chamber was sealed with ZnSe windows to ensure controlled experimental conditions. Before measurement, the sample was degassed at 150 °C for 1 h and subsequently purged with ultrapure nitrogen for 1 h. A mixed gas containing 0.1 mL of methanol was introduced into the chamber, allowing the sample to reach adsorption equilibrium over 30 min before irradiation. The light source used was provided by a xenon lamp coupled through an optical fiber and the light intensity reaching the sample surface was approximately 80 mW cm$^{-2}$. The FT-IR spectra were recorded at 5-min intervals to monitor dynamic changes during the reaction[43].

### Catalysts characterizations

X-ray absorption spectroscopy (XAS) was carried out at the XAS Beamline located within the Australian Synchrotron (ANSTO) in Melbourne, Australia. A collection of Si (111) monochromator crystals, which were maintained at low temperatures using liquid nitrogen, was employed for these measurements. The energy of the electron beam utilized was 3.0 GeV. To reduce the harmonic components of the X-ray beam, optics featuring a silicon-coated collimating mirror and a rhodium-coated focusing mirror were integrated into the beamline. Data collection was performed in transmission mode with W foil used for energy calibration when measuring the W L$_3$ edge, and in fluorescence mode with Ni foil used for energy calibration when measuring the Ni K edge. The size of the beam measured approximately $1 \times 1$ mm$^2$. The XRD patterns were acquired using a Rigaku Rint-2500 diffractometer with Cu Kα radiation, scanning at a speed of 0.1° s$^{-1}$. The morphologies of catalysts were examined using TEM (JEOL 2100) and HRTEM (JEM-3000F). HAADF-STEM images and elemental mapping were performed using a JEOL JEM-2100 F transmission electron microscope. The instrument was equipped with double spherical aberration (Cs) correctors for both the probe-forming and image-forming objective lenses, along with a Super-X EDS detector system. In-situ XPS spectra were obtained using an ESCALAB 250Xi+ analyzer, with Al K$_\alpha$ as the excitation source, and the sample was adsorbed with methanol, dried, and then placed into the reaction chamber, followed by vacuum pumping before testing. The sample was exposed to irradiation using a CEAULIGHT CEL-HXF300 light source, and XPS spectra were collected at 10-min intervals to monitor changes induced by light exposure. The constant analyzer energy mode was used for the acquisition of high-energy resolution spectra with a pass energy of 30 eV and an energy step size of 0.1 eV. UV-Vis-NIR DRS were collected by a UV-Vis-NIR spectrophotometer (JASO V-570). To study the influence of light irradiation on DRS, the powdered samples were firstly irradiated by xenon lamp for different times, and then were moved to UV-Vis-NIR spectrophotometer for DRS measurement. EPR spectra were detected by a Bruker A300 spectrometer. The specific doping metal content was determined by ICP-MS (Agilent 7850 ICP-MS). Raman spectra were collected by Raman microspectroscopy (HORIBA XploRA PLUS). The methanol-TPD patterns were obtained using a chemisorption analyzer (BelCata II, MicrotracBEL, Japan).

## Data availability
Source data are provided in this paper.

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

## Acknowledgements

This work was supported by the National Natural Science Foundation of China (Nos. 22175076 for Z.L., 22222202 for Z. Z., U23A20136 for Z.W. and Z.L., 11904133 for J.L., 22102087 for X.L.), Science and Technology Plan Project of Guangzhou (No. 2023A03J0128), Guangdong S&T Program (2023B1212010008), the Fundamental Research Funds for the Central Universities and Outstanding Innovative Talents Cultivation Funded Programs for Doctoral Students of Jinan University.

## Author contributions

D.T. and Y.L. performed the experiments. X.L., Z.Z., and Z.L. supervised the project and wrote the manuscript. L.M., X.C., Y.C. and X.W. carried out the DFT calculations. J.L., Z.W. C.X., and B.L. analyzed experimental data. All authors discussed the results together and commented on the manuscript.

## Competing interests

The authors declare no competing interests
