## [Transparent Peer Review file · Nature Communications]

Plasmonic Ni-doped $W_{18}O_{49}$ with Dual Active Sites Drives Efficient Methanol Dehydration to Dimethyl Ether

Corresponding Author: Professor Zaizhu Lou

Version 0:

Reviewer comments:

Reviewer #1

(Remarks to the Author)

1. In Figure 2d, the actual content of metal ions in Co-, Pd-, Pt-, Au-, Ag-, and Cu-W18O49 should be mentioned and provided. Whether these properties are optimal methanol dehydration properties for different metal ion doped W18O49?
2. The author mentioned that the enhanced methanol dehydration activity of Nix-W18O49 is due to the construction of C–O bond cleavage and C–O–C coupling. However, the dehydration reaction activity of C2+ alcohols over Nix-W18O49 was also improved a lot, which does not involve a C-O-C coupling process. The authors should give a reasonable explanation.
3. The authors should give an explanation in the manuscript as to why excessive Ni doping in W18O49 leads to a decrease in methanol dehydration activity.
4. In Figure 4j and Supplementary Figure 11, the descriptions of the Raman peaks are differentiated, and the authors should confirm which is correct and name them consistently.
5. In Figure 4m, all the samples exhibit a higher W5+ percentage than W4+ in the dark or under light irradiation. However, in Figure 6a, the percentage of W4+ in Ni0.66- W18O49 is obviously higher than that of W5+ in the methanol atmosphere under light irradiation. The author should recognize the plausibility of this phenomenon and explain it.
6. In Figure 2d, the optimized DME generation rate is 136 mmol/g/h, while 134 mmol/g/h is used in Abstract and Conclusion sections. The author should confirm and correct the numbers.
7. There are some mistakes in References section. In Reference 17, 28, and 34, “Nat. Common” should be corrected to “Nat. Commun”.

Reviewer #2

(Remarks to the Author)

In this paper, the authors present a Ni-doped plasmonic W18O49 nanowire with low-coordinated W-Ni dual active sites for significantly enhanced photocatalytic performances. Starting from theoretical simulations, details of synthetic procedures, systematic characterizations, and various alcohol dehydration reactions are reported. While some research results are interesting, there are quite a lot of specific questions and comments as follows.

1. First of all, using “590-Fold Boost” in the title and anywhere else to get attraction is unfair, because you compare two different reactions under completely different conditions. To be fair, please use the solar-to-chemical conversion efficiency instead.
2. Figure 2b indicates that the DME generation rate over W18O49 is less than 5 mmol g⁻¹ h⁻¹, but in the main text (and in Figure 2d) it shows a different value (14.0 mmol g⁻¹ h⁻¹). What is the true value?
3. In boosting the DME formation, the authors stress the importance of NIR light (compared to UV and Visible light), even almost the same photothermal temperatures are created. The fact is that, judging from the UV-Vis-NIR DRS of W18O49 and Nix-W18O49, there is no specific absorption peak corresponding to NIR. So, what is the intrinsic reason that NIR light is more favorable than UV-Vis light for boosting the DME formation?
4. Judging from the UV-Vis-NIR DRS of W18O49 and Nix-W18O49, it appears that Ni-doping does not contribute extra SPR effect. Why?
5. The authors demonstrated various performances under various light intensities of different light sources. What are the

- respective light-to-chemical conversion efficiencies, especially under concentrated natural solar light irradiation?
6. It is known that the UV-vis-NIR DRS is measured under full-spectrum light irradiation (with a certain range of wavelength). However, the authors showed in Figure 19 (Supporting) that the UV-vis-NIR DRS of Ni_{0.66}-W18O49 varied with light irradiation. It is hard for the reviewer to understand this result. Please explain.
7. What is the beam diameter of the Xe lamp used? The sample was dispersed on an area of as large as ~9.6 cm². How to measure the incident light intensity? How to adjust the average light intensity to be the same when using different optical filters? To the best experience of the reviewer, it is quite questionable for a 300 W xenon lamp to emit a light beam with intensities from less than 400 mW cm⁻² to as high as 1000 mW cm⁻² and even higher.
8. Regarding the EXAFS results, the authors state: "Comparatively, Ni_{0.66}-W18O49 only shows a strong peak at 1.52 Å, attributing to the Ni-O bond, which suggests that Ni is present as an isolated atom in Ni_{0.66}-W18O49." What does "Ni is present as an isolated atom" mean? Ni single atom? If so, don't the doped Ni atoms enter/occupy the lattice of W18O49? If not, how could doping of low valent Ni to induce more low-valent W₄₊/W₅₊?
9. The authors mention that doping of 0.66 wt% Ni in W18O49 facilitates photoelectron trapping preferentially on low-valent W₄₊ sites. Why are the photoelectrons not trapped on the more positive high-valent W₆₊ (and even W₅₊) sites?
10. The DME generation rate over Ni_x-W18O49 increased, but the DME selectivity decreased with increasing the Ni doping amount. Why?
11. In the proposed mechanism, the authors mentioned the occurrence of few intermediates like *CH₃, OH radicals. However, those intermediates were not observed in the in situ FTIR measurement, why?

Given the aforementioned so many questions, I would not recommend acceptance of this manuscript.

Reviewer #3

(Remarks to the Author)

In the submission, the authors constructed Ni-doped plasmonic W18O49 nanowire architecture with exceptional high plasmonic photocatalytic performance in dehydration of methanol to DME and of higher alcohols (>C₂) for alkene. The W18O49 nanowire architecture was reported by the authors as an active plasmonic photocatalyst (for the dehydration of alcohols (Ref. 16-18), and in the present work, the authors further screened a series of doped W18O49 nanowire architectures to acquire the best Ni-W18O49 photocatalysts. Although the photocatalytic performance is impressive, the reaction mechanism is rather ambiguous and the comparisons between photocatalytic and thermocatalytic dehydration reactions are not appropriately justified. Thus I do not consider that the manuscript is suitable for the publication in Nature Communications.

1. The observation of apparent activation energy for the photocatalytic dehydration of methanol to DME suggested the existence of thermal-activated steps. However, the authors failed to discuss this, which made the proposed reaction mechanism doubtful.
2. Energy diagram of methanol dehydration reaction to DME on the Ni-W18O49 catalyst needs to be calculated.
3. Fig. 1 and Fig. 6e: The authors proposed molecularly-adsorbed CH₃OH as the active species for the dehydration reaction, while in the thermocatalytic reactions, the methoxyl species is the active species (one paper I remembered, Angew. Chem. Int. Ed. 2016, 55, 623).
4. The authors are encouraged to compare the photocatalytic performance with the typical thermocatalytic performance in the literature. It is also recommended to estimate which temperatures are needed for the Ni-W18O49 catalyst to thermocatalytically catalyze the CH₃OH dehydration reaction to achieve the photocatalytic performance.
5. The photocatalytic performance varied greatly with the reactant, for example, propanol and isopropanol. Why?
6. How about the carbon balance of the photocatalytic reaction?
7. Did the authors detect water, the other product for the CH₃OH dehydration reaction?
8. How accurately was the sample temperature measured during the photocatalytic reaction to make the Arrhenius plots?
9. Please analyze the charge balance mechanism for Ni²⁺ doped W18O49.
10. The information about the light source in Fig. 1 caption was not complete.
11. Title: "W18O49" should be "Ni-W18O49" since W18O49 did not have dual active sites.

Version 1:

Reviewer comments:

Reviewer #1

(Remarks to the Author)

The authors well addressed the reviewers' concerns, the present form can be accepted for publication.

Reviewer #2

(Remarks to the Author)

The reviewer carefully read the revised manuscript and the responses to reviewers. It appears that the authors have made significant improvements in the revised manuscript by addressing the questions and comments of each reviewer properly. So, I would like to recommend acceptance of the current version of manuscript.

Reviewer #3

(Remarks to the Author)

In the revised submission the authors have improved the manuscript largely. However, the key concerns about the mechanism still hold, particularly the authors failed to rationalize some experimental results using DFT calculations:

1. Figure 6e illustrates the proposed reaction mechanism. The authors need consider to rationalize the following experimental results: 1) why was NIR light illumination more effective in decreasing the experimentally-measured E_a than UV and Vis light illumination (Fig. 4c); 2) why did the experimentally-measured E_a vary with the power of adopted light?
2. Fig. 6b-c: In addition to the C-O-C region, the C-O region also obviously changed, which needs to be discussed; meanwhile, did the formation of DME give any new features in the C 1s XPS spectra? Moreover, both infrared and XPS results did not demonstrate the high methanol conversions shown in Fig. 3a, why?

Version 2:

Reviewer comments:

Reviewer #3

(Remarks to the Author)

In the revised submission the authors replied to the previous comments appropriately and revised the manuscript accordingly. I recommend to accept the revised submission.

Point-by-Point Response to Reviewers

Reviewer: 1

Comments:

1. In Figure 2d, the actual content of metal ions in Co-, Pd-, Pt-, Au-, Ag-, and Cu- $W_{18}O_{49}$ should be mentioned and provided. Whether these properties are optimal methanol dehydration properties for different metal ion doped $W_{18}O_{49}$?

Response: We thank the reviewer for the valuable suggestions. In response, a comprehensive series of comparative photocatalysts M- $W_{18}O_{49}$ (M = Co, Pd, Pt, Au, Ag, and Cu) were synthesized across a range of dopant concentrations. The photocatalytic methanol dehydration performance of these materials was systematically evaluated (Supplementary Figure S6). Notably, the optimal dimethyl ether (DME) production rates achieved over the Co-, Pd-, Pt-, Au-, Ag-, and Cu-doped $W_{18}O_{49}$ samples were 19.6, 30.6, 126.4, 95.7, 67.7, and 43.6 $\text{mmol g}^{-1} \text{h}^{-1}$, respectively. Critically, all of these optimized rates remain significantly lower than the performance of the benchmark $Ni_{0.66}-W_{18}O_{49}$ ($135.9 \text{ mmol g}^{-1} \text{h}^{-1}$, Figure 2d). Inductively coupled plasma mass spectrometry (ICP-MS) analysis confirmed the actual dopant loadings in these optimal M- $W_{18}O_{49}$ samples to be 0.61 wt% (Co), 0.73 wt% (Pd), 1.04 wt% (Pt), 0.96 wt% (Au), 0.75 wt% (Ag), and 0.69 wt% (Cu) (Supplementary Table S2).

Supplementary Figure S6. **a** Photocatalytic DME generation rates over Co_{low}- $W_{18}O_{49}$, Co- $W_{18}O_{49}$, Co_{high}- $W_{18}O_{49}$, Pd_{low}- $W_{18}O_{49}$, Pd- $W_{18}O_{49}$, Pd_{high}- $W_{18}O_{49}$, Pt_{low}- $W_{18}O_{49}$, Pt- $W_{18}O_{49}$, Pt_{high}- $W_{18}O_{49}$, Au_{low}- $W_{18}O_{49}$, Au- $W_{18}O_{49}$, Au_{high}- $W_{18}O_{49}$, Ag_{low}- $W_{18}O_{49}$, Ag- $W_{18}O_{49}$, Ag_{high}- $W_{18}O_{49}$, Cu_{low}- $W_{18}O_{49}$, Cu- $W_{18}O_{49}$, and Cu_{high}- $W_{18}O_{49}$, respectively. **b** Product selectivity of photocatalytic methanol dehydration over $W_{18}O_{49}$, Co- $W_{18}O_{49}$, Pd- $W_{18}O_{49}$, Pt- $W_{18}O_{49}$, Au- $W_{18}O_{49}$, Ag- $W_{18}O_{49}$ and Cu- $W_{18}O_{49}$, respectively.

Supplementary Table 2. ICP-MS measurement results of M-W₁₈O₄₉ (M= Co, Pd, Pt, Au, Ag, and Cu)

Sample	m ₀ (g)	Element	C ₀ (mg/L)	f	C ₁ (mg/L)	C _x (mg/kg)	W (%)
Au-W ₁₈ O ₄₉	0.03077	Au	5.802	1	5.802	9428.01	0.963%
	0.03098		6.0873	1	6.0873	9824.57	
Ag-W ₁₈ O ₄₉	0.03041	Ag	4.5307	1	4.5307	7449.43	0.746%
	0.03138		4.6914	1	4.6914	7475.16	
Cu-W ₁₈ O ₄₉	0.03119	Cu	4.2723	1	4.2723	6848.61	0.687%
	0.03114		4.2864	1	4.2864	6882.51	
Pt-W ₁₈ O ₄₉	0.03084	Pt	1.2773	5	6.3865	10354.25	1.04%
	0.03071		1.2764	5	6.382	10390.75	
Pd-W ₁₈ O ₄₉	0.0305	Pd	0.8954	5	4.477	7339.34	0.73%
	0.03129		0.9145	5	4.5725	7306.65	
Co-W ₁₈ O ₄₉	0.03108	Co	3.7787	1	3.7787	6078.96	0.61%
	0.03082		3.7654	1	3.7654	6108.64	

m₀: Sample weight; V₀: 50 mL; C₀: Element concentration in the test solution; f: Dilution factor; C₁: Element concentration in the original digestion solution; C_x: Elemental content of the sample; W (%): The final test result of the measured element is expressed as a percentage of the total composition. The test result is obtained using the following formula:

$$C_x(\mu\text{g}/\text{kg}) = \frac{C_0(\mu\text{g}/\text{L}) \times f \times V_0(\text{mL}) \times 10^{-3}}{m(\text{g}) \times 10^{-3}} = \frac{C_1(\text{mg}/\text{L}) \times V_0(\text{mL})}{m(\text{g}) \times 10^{-3}}$$

$$W(\%) = \frac{C_x(\text{mg}/\text{kg})}{10^6}$$

Modifications:

- 1) The sentence “For comparative studies, control samples including Co-, Pd-, Pt-, Au-, Ag-, and Cu-W₁₈O₄₉ were also prepared.” was modified to “For comparative studies, control samples including Co-, Pd-, Pt-, Au-, Ag-, and Cu-W₁₈O₄₉ with varying dopant concentrations were also synthesized.” at p5 of the revised manuscript.
- 2) The sentence “The DME production rates of Co-, Pd-, Pt-, Au-, Ag-, and Cu-W₁₈O₄₉ are 16, 29, 104, 89, 65, and 41 mmol g⁻¹ h⁻¹, respectively, which are all lower than that of Ni_{0.66}-W₁₈O₄₉.” was modified to be “The optimal DME production rates of Co-, Pd-, Pt-, Au-, Ag-, and Cu-W₁₈O₄₉ are 19.6, 30.6, 126.4, 95.7, 67.7, and 43.6 mmol g⁻¹ h⁻¹, respectively, which are all lower than that of Ni_{0.66}-W₁₈O₄₉.” at p6 of the revised manuscript.
- 3) The sentence “The actual contents of doped metal ions in the optimal Co-, Pd-, Pt-, Au-, Ag-, and Cu-W₁₈O₄₉ samples were measured by inductively coupled plasma mass spectrometry (ICP-MS, Supplementary Figure Table 2), which are 0.61 wt%, 0.73 wt%, 1.04 wt%, 0.96 wt%, 0.75 wt%, and 0.69 wt%, respectively.” was added at p6 of the revised manuscript.
- 4) Supplementary Figure S6 was modified and Supplementary Table 2 was added in the revised Supplementary Materials.

- 5) **Figure 2d** was updated by the optimal photocatalytic DME generation rates over Co-W₁₈O₄₉, Pd-W₁₈O₄₉, Pt-W₁₈O₄₉, Au-W₁₈O₄₉, Ag-W₁₈O₄₉, and Cu-W₁₈O₄₉.

Figure 2d. The optimal DME generation rates over Pd-W₁₈O₄₉, Pt-W₁₈O₄₉, Co-W₁₈O₄₉, Au-W₁₈O₄₉, Ag-W₁₈O₄₉, and Cu-W₁₈O₄₉.

2. The author mentioned that the enhanced methanol dehydration activity of Ni_x-W₁₈O₄₉ is due to the construction of C–O bond cleavage and C–O–C coupling. However, the dehydration reaction activity of C₂₊ alcohols over Ni_x-W₁₈O₄₉ was also improved a lot, which does not involve a C–O–C coupling process. The authors should give a reasonable explanation.

Response: We thank the reviewer for the good comment. As shown in W 4f XPS spectra of Supplementary Figure 23, both pristine W₁₈O₄₉ and Ni_{0.66}-W₁₈O₄₉ possess abundance of low-valent W ions (W⁴⁺ and W⁵⁺), leading to their high electron density. After light irradiation, the concentrations of W⁴⁺ and W⁵⁺ in pristine W₁₈O₄₉ and Ni_x-W₁₈O₄₉ are greatly increased due to the photoelectron trapping on tungsten oxide, while the W⁴⁺ concentration (14.1%) in Ni_{0.66}-W₁₈O₄₉ is higher than that in W₁₈O₄₉ (11.6%). This result demonstrates that Ni doping (0.66 wt%) in W₁₈O₄₉ is conducive to facilitating the capture of photoelectrons, resulting in more low-valent W⁴⁺. The EPR spectra in Figure 4l and Supplementary Figure 21 further support that light irradiation can increase the electron density of Ni_{0.66}-W₁₈O₄₉ more efficiently than that of pristine W₁₈O₄₉. More surface low-valent W⁴⁺ and W⁵⁺ in Ni_{0.66}-W₁₈O₄₉ can facilitate the adsorption of C₂₊ alcohols, which is crucial for boosting photocatalytic C₂₊ alcohol dehydration.

To further verify the stronger adsorption of Ni_{0.66}-W₁₈O₄₉ than W₁₈O₄₉ for C₂₊ alcohols under light irradiation, we performed *in-situ* FTIR spectroscopy measurements. As shown in Supplementary Figure 33, Ni_{0.66}-W₁₈O₄₉ exhibits stronger ν_{C–O}, δ_{O–H}, δ_{CH₃}, and ν_{CH₃} peaks than pristine W₁₈O₄₉ after UV light irradiation, further suggesting that Ni_{0.66}-W₁₈O₄₉ with more surface low-valent W⁴⁺ can adsorb more C₂₊ alcohols than pristine W₁₈O₄₉ under light irradiation. Consequently, Ni_{0.66}-W₁₈O₄₉ with more light-induced surface low-valent W⁴⁺ and W⁵⁺ can facilitate C₂₊ alcohol adsorption, and its stronger SPR can generate more hot electrons to promote C–O bond cleavage of C₂₊ alcohol for alkene generation.

Supplementary Figure 23. W 4f XPS spectra of $W_{18}O_{49}$ (a), $Ni_{0.66}-W_{18}O_{49}$ (b) and $Ni_{1.4}-W_{18}O_{49}$ (c) before and after light irradiation.

Supplementary Figure 21. EPR spectra of $W_{18}O_{49}$, $Ni_{0.66}-W_{18}O_{49}$ and $Ni_{1.4}-W_{18}O_{49}$ before (dotted line) and after light irradiation (solid line).

Supplementary Figure 33. In-situ FT-IR spectroscopy of (a) $W_{18}O_{49}$ and (b) $Ni_{0.66}-W_{18}O_{49}$ during photocatalytic tert-butanol dehydration reaction under UV light irradiation, and (c) $Ni_{0.66}-W_{18}O_{49}$ under UV for 5 min and NIR for 15 min.

Modifications:

- 1) The sentence “Figure 4I and Supplementary Figure 16 show that $Ni_{0.66}-W_{18}O_{49}$ has a significantly increased electron density after light irradiation, suggesting that an appropriate amount of Ni-doping facilitates photoelectron accumulation on $Ni_{0.66}-W_{18}O_{49}$.” was modified to be “Electron paramagnetic resonance (EPR) spectra (Figure 4I and Supplementary Figure 21) show that $Ni_{0.66}-W_{18}O_{49}$ has a significantly increased electron density after light irradiation, suggesting that an appropriate amount of Ni-doping facilitates photoelectron accumulation on $Ni_{0.66}-W_{18}O_{49}$.” at p13 of the revised manuscript.
- 2) The sentences “The reaction pathway of C_{2+} alcohol (tert-butanol) dehydration over $W_{18}O_{49}$ and $Ni_{0.66}-W_{18}O_{49}$ was monitored by the in-situ FT-IR spectroscopy as shown in Supplementary Figure 33. UV-irradiation can promote more tert-butanol adsorption on $Ni_{0.66}-W_{18}O_{49}$, and NIR-irradiation promotes C-O bond cleavage for C=C bond generation.” were added at p17 of the revised manuscript.
- 3) The sentences “While, for C_{2+} alcohol dehydration, $Ni_{0.66}-W_{18}O_{49}$ with more light-induced surface low-valent W^{4+} and W^{5+} can facilitate C_{2+} alcohol adsorption, meanwhile, its hot electrons promote C-O bond cleavage for alkene generation.” was added at p20 of the revised manuscript.
- 4) **Supplementary Figure 33** was added in the revised Supplementary Materials.

3. The authors should give an explanation in the manuscript as to why excessive Ni doping in $W_{18}O_{49}$ leads to a decrease in methanol dehydration activity.

Response: We thank the reviewer for the valuable suggestions. As shown in the FT-EXAFS spectra (Figure 5b), $Ni_{0.66}-W_{18}O_{49}$ only shows a strong peak at 1.52 Å, attributed to the Ni-O bond, which suggests that Ni is atomically dispersed in $Ni_{0.66}-W_{18}O_{49}$. However, $Ni_{1.4}-W_{18}O_{49}$ exhibits two peaks of similar intensity at 1.52 Å and 2.62 Å, corresponding to the Ni-O bond of atomically dispersed Ni atoms and the Ni-Ni bond characteristic of NiO, respectively, which implies that Ni in $Ni_{1.4}-W_{18}O_{49}$ is not present as single atom doping. Compared to $Ni_{1.4}-W_{18}O_{49}$, $Ni_{0.66}-W_{18}O_{49}$ with atomically dispersed Ni atoms can expose more Ni-W dual active sites, which are crucial for driving the methanol-to-DME conversion pathway. Therefore, excessive Ni doping in $W_{18}O_{49}$ ($Ni_{1.4}-W_{18}O_{49}$) leads to a decrease in photocatalytic methanol dehydration activity. Moreover, the

EPR spectra of $\text{Ni}_{0.66}\text{-W}_{18}\text{O}_{49}$ and $\text{Ni}_{1.4}\text{-W}_{18}\text{O}_{49}$ (Supplementary Figure 21) manifest that light irradiation can increase the electron density of $\text{Ni}_{0.66}\text{-W}_{18}\text{O}_{49}$ more efficiently than that of $\text{Ni}_{1.4}\text{-W}_{18}\text{O}_{49}$. The relatively higher SPR intensity of $\text{Ni}_{0.66}\text{-W}_{18}\text{O}_{49}$ than $\text{Ni}_{1.4}\text{-W}_{18}\text{O}_{49}$ under light irradiation is favorable to photocatalytic methanol dehydration. In addition, the temperature-programmed desorption (TPD) profile of methanol on $\text{Ni}_{1.4}\text{-W}_{18}\text{O}_{49}$ was measured as shown in Supplementary Figure 34, and excessive Ni doping in $\text{Ni}_{1.4}\text{-W}_{18}\text{O}_{49}$ reduces the adsorption amount of methanol, restraining the dehydration reaction for DME generation.

Figure 5b. FT-EXAFS spectra of $\text{Ni}_{0.66}\text{-W}_{18}\text{O}_{49}$, $\text{Ni}_{1.4}\text{-W}_{18}\text{O}_{49}$, NiO, and Ni foil references.

Supplementary Figure 21. EPR spectra of $\text{W}_{18}\text{O}_{49}$, $\text{Ni}_{0.66}\text{-W}_{18}\text{O}_{49}$ and $\text{Ni}_{1.4}\text{-W}_{18}\text{O}_{49}$ before (dotted line) and after light irradiation (solid line).

Supplementary Figure 34. The methanol-TPD pattern of the W₁₈O₄₉, Ni_{0.66}-W₁₈O₄₉ and Ni_{1.4}-W₁₈O₄₉ catalysts.

Modifications: .

- 1) The sentence “While excessive Ni doping reduces the adsorption amount of methanol, restraining the dehydration reaction for DME generation (Supplementary Figure 34).” was added at p17 of the revised manuscript.
- 2) **Supplementary Figure 34** was added in the revised Supplementary Materials.

4. In Figure 4j and Supplementary Figure 11, the descriptions of the Raman peaks are differentiated, and the authors should confirm which is correct and name them consistently.

Response: We have revised the description of the Raman peaks in Supplementary Figure 14.

Supplementary Figure 14. Raman spectra of samples W₁₈O₄₉ and Ni_x-W₁₈O₄₉ (x=0.15, 0.32, 0.66, 1.0, and 1.4).

5. In Figure 4m, all the samples exhibit a higher W^{5+} percentage than W^{4+} in the dark or under light irradiation. However, in Figure 6a, the percentage of W^{4+} in $Ni_{0.66}-W_{18}O_{49}$ is obviously higher than that of W^{5+} in the methanol atmosphere under light irradiation. The author should recognize the plausibility of this phenomenon and explain it.

Response: We thank the reviewer for the valuable comment. Due to the electron-donating nature of methanol, the W 4f XPS peak has a shift to lower binding energy after adsorbed methanol, as displayed in Figure 6a. Under light irradiation, the photoelectrons on $Ni_{0.66}-W_{18}O_{49}$ are trapped by the high-valent W^{6+} , resulting in more W^{5+} and W^{4+} . More content of W^{4+} ions is formed due to the reducing environment in the methanol atmosphere.

Modifications:

- 1) The sentence “The increase in W^{4+} content is attributed to photoelectron accumulation on $Ni_{0.66}-W_{18}O_{49}$ ” was modified to be “The increase in W^{4+} content is attributed to photoelectron accumulation on $Ni_{0.66}-W_{18}O_{49}$ in the methanol atmosphere.” at p16 of the revised manuscript.

6. In Figure 2d, the optimized DME generation rate is 136 mmol/g/h, while 134 mmol/g/h is used in Abstract and Conclusion sections. The author should confirm and correct the numbers.

Response: The optimized DME generation rate should be $136 \text{ mmol g}^{-1} \text{ h}^{-1}$, and we have revised the related descriptions in Abstract and Conclusion sections.

7. There are some mistakes in References section. In Reference 17, 28, and 34, “Nat. Common” should be corrected to “Nat. Commun”.

Response: We have corrected the “Nat. Common” to be “Nat. Commun.” in the References section. Moreover, the names of the cited Journals have been checked carefully in all References.

Reviewer: 2

Comments:

In this paper, the authors present a Ni-doped plasmonic $W_{18}O_{49}$ nanowire with low-coordinated W-Ni dual active sites for significantly enhanced photocatalytic performances. Starting from theoretical simulations, details of synthetic procedures, systematic characterizations, and various alcohol dehydration reactions are reported. While some research results are interesting, there are quite a lot of specific questions and comments as follows.

1. First of all, using “590-Fold Boost” in the title and anywhere else to get attraction is unfair, because you compare two different reactions under completely different conditions. To be fair, please use the solar-to-chemical conversion efficiency instead.

Response: Thanks for the reviewer’s suggestion. In our experiment, $Ni_{0.66}-W_{18}O_{49}$ generated $136 \text{ mmol g}^{-1} \text{ h}^{-1}$ DME from methanol dehydration under full-spectrum irradiation (400 mW cm^{-2}), and the surface temperature of the catalyst was measured to be $108 \text{ }^\circ\text{C}$. For comparison, the thermocatalytic DME generation rate from methanol dehydration was measured to be 0.23 mmol

$\text{g}^{-1} \text{h}^{-1}$ at $108\text{ }^{\circ}\text{C}$. Under the same reaction temperature of $108\text{ }^{\circ}\text{C}$, the photocatalytic performance of $\text{Ni}_{0.66}\text{-W}_{18}\text{O}_{49}$ is 590-fold higher than its thermocatalytic performance.

The energy changes in methanol dehydration to dimethyl ether (DME) involve enthalpy (ΔH), Gibbs free energy (ΔG), and activation energy (E_a), and each of them are critical for understanding thermodynamics and kinetics. Below is a concise analysis:

Key Energy Changes:

Standard ΔH is -23.9 kJ mol^{-1} at $25\text{ }^{\circ}\text{C}$, indicating an exothermic reaction.

Standard ΔG is -17 kJ mol^{-1} at $25\text{ }^{\circ}\text{C}$, indicating a thermodynamic spontaneous reaction.

Activation Energy (E_a) is high as $\sim 200\text{ kJ mol}^{-1}$ without catalysts.

As an exothermic reaction, methanol dehydration does not primarily utilize light energy for chemical energy conversion. Actually, the activation energy (E_a) is critical in limiting the kinetics of methanol dehydration reaction. In the absence of catalysts, the E_a is as high as 200 kJ mol^{-1} , but this value decreases to $100\text{--}130\text{ kJ mol}^{-1}$ with $\gamma\text{-Al}_2\text{O}_3$ catalysts. In our experiment, light irradiation excites surface plasmon resonance (SPR) of $\text{Ni}_{0.66}\text{-W}_{18}\text{O}_{49}$, generating hot electrons that promote C-O bond cleavage of methanol. This process greatly reduces the E_a of methanol dehydration, thereby boosting DME generation. Under light intensities of $200, 400, 600, 800,$ and 1000 mW cm^{-2} , the measured E_a values drop to $81.3, 66.4, 48.3, 33.9,$ and 23.4 kJ mol^{-1} , respectively, which are all far lower than the thermocatalytic E_a of 142.4 kJ mol^{-1} . Consequently, the light energy is mainly used to excite hot electrons, which drastically lower the E_a and boost methanol dehydration for DME generation. To avoid misunderstanding, we have removed the term “590-fold” from the title of the revised manuscript.

Modifications:

- 1) The title “Semiconductor-Plasmon Drives Dual Active Sites of $\text{W}_{18}\text{O}_{49}$ for a 590-Fold Boost in Dimethyl Ether Generation” was modified to be “Semiconductor-Plasmon Drives Dual Active Sites of Ni-doped $\text{W}_{18}\text{O}_{49}$ for Efficient Dimethyl Ether Generation via Methanol Dehydration” at p1 of the revised manuscript.
- 2) The sentence “As an exothermic reaction ($\Delta H, -23.9\text{ kJ mol}^{-1}$), the light energy is mainly used to reduce the E_a of methanol dehydration and promote the kinetics of DME generation.” was added at p10 of the revised manuscript.

2. Figure 2b indicates that the DME generation rate over $\text{W}_{18}\text{O}_{49}$ is less than $5\text{ mmol g}^{-1} \text{h}^{-1}$, but in the main text (and in Figure 2d) it shows a different value ($14.0\text{ mmol g}^{-1} \text{h}^{-1}$). What is the true value?

Response: The true DME generation rate over $\text{W}_{18}\text{O}_{49}$ is $14.0\text{ mmol g}^{-1} \text{h}^{-1}$. In order to avoid the misunderstanding, the left vertical axis was exchanged with the right one in **Figure 2b** of the revised manuscript.

Figure 2b. The DME generation rate and product selectivity over $Ni_x-W_{18}O_{49}$.

3. In boosting the DME formation, the authors stress the importance of NIR light (compared to UV and Visible light), even almost the same photothermal temperatures are created. The fact is that, judging from the UV-Vis-NIR DRS of $W_{18}O_{49}$ and $Ni_x-W_{18}O_{49}$, there is no specific absorption peak corresponding to NIR. So, what is the intrinsic reason that NIR light is more favorable than UV-Vis light for boosting the DME formation?

Response: We thank the reviewer for the valuable comments. The temperature variation over $Ni_{0.66}W_{18}O_{49}$ under different light irradiations (UV, Vis, UV-Vis, NIR, Vis-NIR and UV-Vis-NIR with a fixed intensity of 400 mW cm^{-2}) was measured as shown in Figure 3b, and NIR, Vis-NIR and UV-Vis-NIR irradiations could increase the surface temperature to around $108 \text{ }^\circ\text{C}$. It demonstrates that the SPR in the NIR region makes a dominant contribution to the photothermal effect. Meanwhile, the DME generation rate of $Ni_{0.66}W_{18}O_{49}$ under NIR light ($105 \text{ mmol g}^{-1} \text{ h}^{-1}$) is far better than that under UV light ($0.8 \text{ mmol g}^{-1} \text{ h}^{-1}$), Vis light ($3.4 \text{ mmol g}^{-1} \text{ h}^{-1}$), or UV-Vis light ($35 \text{ mmol g}^{-1} \text{ h}^{-1}$), suggesting that the strong SPR of $Ni_{0.66}W_{18}O_{49}$ in the NIR region plays a vital role in photocatalytic methanol dehydration for DME generation. While the thermocatalytic methanol dehydration just generates $0.23 \text{ mmol g}^{-1} \text{ h}^{-1}$ DME at $108 \text{ }^\circ\text{C}$. These results demonstrate that the photothermal effect is not the dominant contribution to enhancing methanol dehydration. Besides the photothermal effect, the SPR of $Ni_{0.66}W_{18}O_{49}$ can generate hot electrons which can promote the C-O bond cleavage and C-O-C coupling on low-valent W and Ni sites, boosting DME generation. This process has been demonstrated by in-situ C 1s XPS and in-situ FT-IR spectra in Figure 6a and Supplementary Figure 32.

Figure 3b. The surface temperature of Ni_{0.66}-W₁₈O₄₉ and its photocatalytic DME generation rates under different light irradiations (UV, Vis, UV-Vis, NIR, Vis-NIR, and UV-Vis-NIR, with a fixed intensity of 400 mW cm⁻²).

Figure 6a. In-situ C 1s and W 4f XPS spectra of Ni_{0.66}-W₁₈O₄₉ during photocatalytic methanol dehydration reaction.

Supplementary Figure 32. *In-situ* FT-IR transmission spectra of Ni_{0.66}-W₁₈O₄₉ during photocatalytic methanol dehydration reaction under UV irradiation for 5 min, and then NIR irradiation for 15 min.

From the UV-Vis-NIR DRS in Figure 4k, both W₁₈O₄₉ and Ni_x-W₁₈O₄₉ display strong SPR absorption across the Vis-NIR region (from 420-1400 nm). To precisely identify the SPR

absorption characteristics, we further measured the UV-Vis-NIR DRS of $\text{Ni}_{0.66}\text{-W}_{18}\text{O}_{49}$ over an extended wavelength range (200–2500 nm). As shown in Supplementary Figure 19, $\text{Ni}_{0.66}\text{-W}_{18}\text{O}_{49}$ exhibits broad light absorption spanning 420–2500 nm, with a distinct SPR absorption peak centered at 900 nm. Because UV-Vis-NIR DRS is collected from the powders of samples, and the coupling between different particles results in a broad band of SPR in the spectra. For a better understanding of the SPR of $\text{W}_{18}\text{O}_{49}$, the SPR band of a single $\text{W}_{18}\text{O}_{49}$ nanowire was simulated by FDTD calculation as shown in Supplementary Figure 20. One clear band with peak around 900-1000 is observed for a single nanowire, demonstrating its character of SPR. However, for various nanowires with different lengths and diameters, their calculated SPR band becomes broad in the NIR region, demonstrating that the broad SPR band in UV-Vis-NIR DRS is due to the coupling between different nanowires.

Figure 4k. UV-Vis-NIR DRS of $\text{W}_{18}\text{O}_{49}$ and $\text{Ni}_x\text{-W}_{18}\text{O}_{49}$.

Supplementary Figure 19. UV-Vis-NIR DRS of $\text{Ni}_{0.66}\text{-W}_{18}\text{O}_{49}$.

Supplementary Figure 20. Electric field distribution and light absorption spectra of single $W_{18}O_{49}$ nanowire (a, b) and various nanowires (c, d), by FDTD simulations.

The $Ni_{0.66}W_{18}O_{49}$ with strong SPR and abundant low-valent W and Ni sites can adsorb methanol molecules, meanwhile, the NIR-excited hot electrons facilitate the C-O bond cleavage and C-O-C coupling by reducing the E_a to 71.8 kJ mol⁻¹. However, UV- and Vis-excited photoelectrons just reduce E_a to 127.9 and 119 kJ mol⁻¹, respectively. The E_a of thermocatalytic methanol dehydration over $Ni_{0.66}W_{18}O_{49}$ is 142.4 kJ mol⁻¹. Therefore, NIR-irradiation is more favorable than UV- and Vis-irradiation in promoting methanol dehydration for DME generation.

Modifications:

- 1) The sentence “Although the SPR intensity of $Ni_x-W_{18}O_{49}$ experiences a slight reduction, the overall SPR characteristics remain robust (Figure 4k)” was modified to be “Although the SPR intensity of $Ni_x-W_{18}O_{49}$ experiences a slight reduction, the overall SPR characteristics remain robust (Figure 4k, Supplementary Figure 19)” at p13 of the revised manuscript.
- 2) The sentence “The broad SPR band of $W_{18}O_{49}$ nanowires is due to the coupling of nanowires, demonstrated by FDTD simulations (Supplementary Figure 20).” was added at p13 of the revised manuscript.
- 3) The sentence “When UV light is irradiated on $Ni_{0.66}W_{18}O_{49}$, no vC-O-C band is observed, but as the light switches to NIR, the vC-O-C band appears and strengthens with irradiation time (Supplementary Figure 27), indicating the importance of NIR for DME generation.” was modified to be “When UV light is irradiated on $Ni_{0.66}W_{18}O_{49}$, no vC-O-C band is observed, but as the light switches to NIR, the vC-O-C band appears and strengthens with prolonged irradiation time (Supplementary Figure 32), indicating that NIR-excited plasmonic hot electrons boost DME generation.” at p17 of the revised manuscript.
- 4) Supplementary Figure 19 and Supplementary Figure 20 were added in the revised Supplementary Materials.

4. Judging from the UV-Vis-NIR DRS of $W_{18}O_{49}$ and $Ni_x-W_{18}O_{49}$, it appears that Ni-doping does

not contribute extra SPR effect. Why?

Response: Because Ni^{2+} ions cannot attribute more electron to $\text{W}_{18}\text{O}_{49}$, Ni-doping indeed does not contribute extra SPR effect and even slightly reduces the SPR absorption intensity of $\text{W}_{18}\text{O}_{49}$, as shown in their UV-Vis-NIR DRS (Figure 4k). The SPR enhancement of $\text{Ni}_{0.66}\text{-W}_{18}\text{O}_{49}$ we mentioned in the manuscript refers to the fact that upon light irradiation, $\text{Ni}_{0.66}\text{-W}_{18}\text{O}_{49}$ is more likely to accumulate photogenerated electrons than pristine $\text{W}_{18}\text{O}_{49}$, resulting in a stronger free electron density and SPR absorption intensity. EPR spectra in Supplementary Figure 21 clearly show that $\text{Ni}_{0.66}\text{-W}_{18}\text{O}_{49}$ has more electron density increase than pristine $\text{W}_{18}\text{O}_{49}$ after light irradiation, which leads to a stronger SPR effect for $\text{Ni}_{0.66}\text{-W}_{18}\text{O}_{49}$. The light-enhanced SPR band of $\text{Ni}_{0.66}\text{-W}_{18}\text{O}_{49}$ was further demonstrated from the UV-Vis-NIR DRS as shown in Supplementary Figure 24.

Figure 4k. UV-Vis-NIR DRS of $\text{W}_{18}\text{O}_{49}$ and $\text{Ni}_x\text{-W}_{18}\text{O}_{49}$.

Supplementary Figure 21. EPR spectra of $\text{W}_{18}\text{O}_{49}$, $\text{Ni}_{0.66}\text{-W}_{18}\text{O}_{49}$ and $\text{Ni}_{1.4}\text{-W}_{18}\text{O}_{49}$ before (dotted line) and after light irradiation (solid line).

Supplementary Figure 24. UV-Vis-NIR DRS varies of $W_{18}O_{49}$ and $Ni_{0.66}-W_{18}O_{49}$ under light irradiation.

Modifications:

- 1) The sentence “The light irradiation-enhanced SPR on $Ni_{0.66}-W_{18}O_{49}$ is further confirmed by the increased intensities in their UV-Vis-NIR diffuse reflectance spectra (DRS), as shown in Supplementary Figure 19” was modified to be “The SPR enhancement under light irradiation is more pronounced for $Ni_{0.66}-W_{18}O_{49}$ compared to pristine $W_{18}O_{49}$, as evidenced by their enhanced UV-Vis-NIR diffuse reflectance spectra (DRS), as shown in Supplementary Figure 24.” at p13 of the revised manuscript.
- 2) **Supplementary Figure 24** was updated in the Supplementary Materials.

5. The authors demonstrated various performances under various light intensities of different light sources. What is the respective light-to-chemical conversion efficiencies, especially under concentrated natural solar light irradiation?

Response: We thank the reviewer for the insightful suggestion. The energy changes for methanol dehydration to dimethyl ether (DME) involve enthalpy (ΔH), Gibbs free energy (ΔG), and activation energy (E_a), and each of them is critical for understanding thermodynamics and kinetics. Below is a concise analysis:

Key Energy Changes:

Standard ΔH is $-23.9 \text{ kJ mol}^{-1}$ at $25 \text{ }^\circ\text{C}$, indicating an exothermic reaction.

Standard ΔG is -17 kJ mol^{-1} at $25 \text{ }^\circ\text{C}$, indicating a thermodynamic spontaneous reaction (therm).

Activation Energy (E_a) is high as $\sim 200 \text{ kJ mol}^{-1}$ without catalysts.

As an exothermic reaction, methanol dehydration does not primarily utilize light energy for chemical energy conversion. Actually, the activation energy (E_a) is critical in limiting the kinetics of the methanol dehydration reaction. In the absence of catalysts, the E_a is as high as 200 kJ mol^{-1} , but this value decreases to $100\text{--}130 \text{ kJ mol}^{-1}$ with $\gamma\text{-Al}_2\text{O}_3$ catalysts. In our experiment, light irradiation excites surface plasmon resonance (SPR) of $Ni_{0.66}-W_{18}O_{49}$, generating hot electrons that promote C-O bond cleavage of methanol. This process greatly reduces the E_a of

methanol dehydration, thereby boosting DME generation. Under light intensities of 200, 400, 600, 800, and 1000 mW cm⁻², the measured E_a values drop to 81.3, 66.4, 48.3, 33.9, and 23.4 kJ mol⁻¹, respectively, which are all far lower than the thermocatalytic E_a of 142.4 kJ mol⁻¹. Consequently, the light energy is mainly used to excite hot electrons, which drastically lower the E_a and boost methanol dehydration for DME generation.

Modifications:

- 1) The sentence “As an exothermic reaction (ΔH , -23.9 kJ mol⁻¹), the light energy is mainly used to reduce the E_a of methanol dehydration and promote kinetics of DME generation.” was added at p10 of the revised manuscript.

6. It is known that the UV-vis-NIR DRS is measured under full-spectrum light irradiation (with a certain range of wavelength). However, the authors showed in Figure 19 (Supporting) that the UV-vis-NIR DRS of Ni_{0.66}-W₁₈O₄₉ varied with light irradiation. It is hard for the reviewer to understand this result. Please explain.

Response: UV-Vis-NIR DRS of powdered Ni_{0.66}-W₁₈O₄₉ samples was collected by a UV-Vis-NIR spectrophotometer (JASCO V-570). The JASCO V-570's light source system combines a deuterium lamp (UV) + halogen lamp (Vis-NIR) with automatic switching technology (325–370 nm transition), a monochromator system, enabling high-precision optical measurements across the full spectrum (185–2500 nm). So, DRS of samples were collected by various monochromatic lights in the UV-Vis-NIR region produced by the monochromator in the spectrophotometer. Due to the weak intensity of monochromatic light, the UV-Vis-NIR DRS of Ni_{0.66}-W₁₈O₄₉ does not have remarkable changes. While, in our experiment, the powdered samples were firstly irradiated under UV-Vis-NIR light (400 mW cm⁻²) supplied by a xenon lamp for different times. Then, the samples were moved to a spectrophotometer (JASCO V-570) for collecting UV-Vis-NIR DRS.

Modifications:

- 1) Sentence “To study the influence of light irradiation on DRS, the powdered samples were firstly irradiated by xenon lamp for different times, and then were moved to UV-Vis-NIR spectrophotometer for DRS measurement” was added in p24 of the revised manuscript.

7. What is the beam diameter of the Xe lamp used? The sample was dispersed on an area of as large as ~9.6 cm². How to measure the incident light intensity? How to adjust the average light intensity to be the same when using different optical filters? To the best experience of the reviewer, it is quite questionable for a 300 W xenon lamp to emit a light beam with intensities from less than 400 mW cm⁻² to as high as 1000 mW cm⁻² and even higher.

Response: In our experiment, the beam diameter in the output of the Xe lamp is 53 mm. The beam size will become larger as the distance increases to the output, and the beam diameter used in our experiment is 100 mm. The samples were coated on a cover glass with an area of

9.6 cm², which is smaller than the beam size. The incident light intensity is measured by using an optical power meter (PL-MW 2000 Photoradiometer, PerfectLight). We can control the working current of the Xenon lamp to adjust the average light intensity to 400 mW cm⁻². As show in following figures, we adjusted the working current to be 16.3, 13.5, 21.0, 13.0, 14.2, and 12.5 A for UV (200-400 nm), Vis (400-780 nm), NIR (>800), UV-Vis (200-780nm), Vis-NIR (>400nm), and UV-Vis-NIR (>200 nm) light to obtain the same intensity of 400 mW cm⁻², respectively. To measure the incident light intensity more accurately, we measured the light intensity at two different locations (Figures). Based on the measured light intensity at the two locations and the total light area of 9.6 cm², we can roughly get the average light intensity irradiated on the samples. The working currents of the Xenon lamp can be adjusted to 11.0, 12.6, 15.4, 18.1, and 20.5 A to obtain 200, 400, 600, 800, and 1000 mW cm⁻² UV-Vis-NIR light, respectively. The maximum intensity of UV-Vis-NIR light supplied by the 300 W Xe lamp (Perfectlight, PLS-SXE300D) can reach 1300 mW cm⁻².

Figures of light beam and samples coated on cover glass. Red circle is size (9.6 cm²) of samples coated cover glass, and yellow circle is the size (3.14 cm²) of detector of optical power meter.

Figures of instrument settings to obtain different light irradiation with a constant intensity of 400 mW cm⁻².

Figures of instrument settings to obtain different intensities of UV-Vis-NIR light irradiation.

Modifications:

- 1) The sentence “The light intensity can be adjusted by modulating the working current of light source.” at p21 of the revised manuscript.

8.Regarding the EXAFS results, the authors state: “Comparatively, $\text{Ni}_{0.66}\text{-W}_{18}\text{O}_{49}$ only shows a strong peak at 1.52 Å, attributing to the Ni-O bond, which suggests that Ni is present as an isolated atom in $\text{Ni}_{0.66}\text{-W}_{18}\text{O}_{49}$.” What does “Ni is present as an isolated atom” mean? Ni single atom? If so, don’t the doped Ni atoms enter/occupy the lattice of $\text{W}_{18}\text{O}_{49}$? If not, how could doping of low valent Ni to induce more low-valent $\text{W}^{4+}/\text{W}^{5+}$?

Response: Thanks for the reviewer’s valuable comment and kind reminder. We have revised the description in the revised manuscript, and Ni is atomically dispersed within $\text{Ni}_{0.66}\text{-W}_{18}\text{O}_{49}$. During the one-step solvothermal synthesis, Ni was doped into the $\text{W}_{18}\text{O}_{49}$ lattice as single atoms by substituting part of W atoms.

As shown in Supplementary Figure 16, the atom percentages of W^{5+} and W^{4+} in $\text{W}_{18}\text{O}_{49}$, $\text{Ni}_{0.66}\text{-W}_{18}\text{O}_{49}$, and $\text{Ni}_{1.4}\text{-W}_{18}\text{O}_{49}$ reveal no notable variation (before light irradiation), indicating that doping a small amount of Ni did not substantially alter the low-valent W species in the $\text{W}_{18}\text{O}_{49}$ matrix. However, after light irradiation, both the atom percentages of W^{5+} and W^{4+} in $\text{W}_{18}\text{O}_{49}$, $\text{Ni}_{0.66}\text{-W}_{18}\text{O}_{49}$, and $\text{Ni}_{1.4}\text{-W}_{18}\text{O}_{49}$ show significant increases due to the capture of photogenerated electrons. In Supplementary Figure 23, after light irradiation, the W^{5+} content in $\text{W}_{18}\text{O}_{49}$, $\text{Ni}_{0.66}\text{-W}_{18}\text{O}_{49}$, and $\text{Ni}_{1.4}\text{-W}_{18}\text{O}_{49}$ similarly increases from 8.0% to 15.1%, while $\text{Ni}_{0.66}\text{-W}_{18}\text{O}_{49}$ exhibits the highest W^{4+} content of 14.1% than those of $\text{W}_{18}\text{O}_{49}$ (11.6%) and $\text{Ni}_{1.4}\text{-W}_{18}\text{O}_{49}$ (12.4%). This

result demonstrates that doping of 0.66 wt% Ni in $W_{18}O_{49}$ facilitates the capture of photogenerated electrons, thus inducing more low-valent W^{4+}/W^{5+} . In summary, Ni doping itself does not cause a change in the low-valent W^{4+}/W^{5+} ratios within the $W_{18}O_{49}$, but rather Ni doping into the $W_{18}O_{49}$ promotes photogenerated electron trapping, causing significant increases in the low-valent W^{4+}/W^{5+} ratios.

Supplementary Figure 16. XPS of elements W 4f of samples $W_{18}O_{49}$, $Ni_{0.66}-W_{18}O_{49}$ and $Ni_{1.4}-W_{18}O_{49}$.

Supplementary Figure 23. W 4f XPS spectra of $W_{18}O_{49}$ (a), $Ni_{0.66}-W_{18}O_{49}$ (b) and $Ni_{1.4}-W_{18}O_{49}$ (c) before and after light irradiation.

Modifications:

- 1) The sentences “Comparatively, $\text{Ni}_{0.66}\text{-W}_{18}\text{O}_{49}$ only shows a strong peak at 1.52 Å, attributing to the Ni-O bond, which suggests that Ni is present as an isolated atom in $\text{Ni}_{0.66}\text{-W}_{18}\text{O}_{49}$. However, $\text{Ni}_{1.4}\text{-W}_{18}\text{O}_{49}$ exhibits two peaks of similar intensity at 1.52 and 2.62 Å, corresponding to the Ni-O bond of isolated Ni atoms and the Ni-Ni bond characteristic of NiO, [29,30] respectively, which implies that Ni in $\text{Ni}_{1.4}\text{-W}_{18}\text{O}_{49}$ exists in two distinct coordination environments.” were modified to be “Comparatively, $\text{Ni}_{0.66}\text{-W}_{18}\text{O}_{49}$ only shows a strong peak at 1.52 Å, attributed to the Ni-O bond, which suggests that Ni is atomically dispersed within $\text{Ni}_{0.66}\text{-W}_{18}\text{O}_{49}$. However, $\text{Ni}_{1.4}\text{-W}_{18}\text{O}_{49}$ exhibits two peaks of similar intensity at 1.52 and 2.62 Å, corresponding to the Ni-O bond of single Ni atom doping and the Ni-Ni bond characteristic of NiO, [28,29] respectively, which implies that Ni exists in two distinct coordination environments in $\text{Ni}_{1.4}\text{-W}_{18}\text{O}_{49}$.” at p14 of the revised manuscript.

9. The authors mention that doping of 0.66 wt% Ni in $\text{W}_{18}\text{O}_{49}$ facilitates photoelectron trapping preferentially on low-valent W^{4+} sites. Why are the photoelectrons not trapped on the more positive high-valent W^{6+} (and even W^{5+}) sites?

Response: We appreciate the reviewer’s comment, and we acknowledge the inaccuracy in our original description. Under light irradiation, the photogenerated electrons of $\text{Ni}_{0.66}\text{-W}_{18}\text{O}_{49}$ can be trapped on high-valent W^{6+} and even W^{5+} sites to generate more low-valent W^{5+} and W^{4+} , thus enhancing electron density in $\text{Ni}_{0.66}\text{-W}_{18}\text{O}_{49}$ for strong SPR. After light irradiation, $\text{Ni}_{0.66}\text{-W}_{18}\text{O}_{49}$ exhibits the highest W^{4+} content of 14.1% than those of $\text{W}_{18}\text{O}_{49}$ (11.6%) and $\text{Ni}_{1.4}\text{-W}_{18}\text{O}_{49}$ (12.4%), which is favorable to the enhancement of SPR and the adsorption of alcohols.

Modifications:

- 1) The sentence “This result demonstrates that doping of 0.66 wt% Ni in $\text{W}_{18}\text{O}_{49}$ facilitates photoelectron trapping on low-valent W^{4+} sites, thereby enhancing electron density for strong SPR.” was modified to be “This result demonstrates that doping of 0.66 wt% Ni in $\text{W}_{18}\text{O}_{49}$ facilitates photoelectron trapping by high-valent W^{6+} or W^{5+} and generates more low-valent W^{4+} sites, thereby enhancing electron density for strong SPR.” at p13 of the revised manuscript.

10. The DME generation rate over $\text{Ni}_x\text{-W}_{18}\text{O}_{49}$ increased, but the DME selectivity decreased with increasing the Ni doping amount. Why?

Response: As shown in Figure 2b, the selectivity of DME is 99.8%, 98.8%, 98.3%, 98.5%, 98% and 97.6% for $\text{W}_{18}\text{O}_{49}$, $\text{Ni}_{0.15}\text{-W}_{18}\text{O}_{49}$, $\text{Ni}_{0.32}\text{-W}_{18}\text{O}_{49}$, $\text{Ni}_{0.66}\text{-W}_{18}\text{O}_{49}$, $\text{Ni}_{1.0}\text{-W}_{18}\text{O}_{49}$ and $\text{Ni}_{1.4}\text{-W}_{18}\text{O}_{49}$, respectively. The byproducts are detected as CO , CH_4 , and H_2 . The ratios of CH_4 byproduct increase from 0.06% to 1.9% as the doped Ni increases, indicating that the small part of Ni sites promotes CH_4 generation. During methanol dehydration, plasmonic hot electrons can facilitate the C-O bond cleavage of methanol on low-valent W sites, generating $^*\text{CH}_3$ as intermediates. Meanwhile, $^*\text{CH}_3$ can react with methanol absorbed on Ni sites to facilitate C-O-C coupling and DME generation, and this process happens rapidly. While, small part of doped Ni can act as active sites for hydrogen generation. Therefore, very little amount of $^*\text{CH}_3$ can react with the $^*\text{H}$

on Ni sites for CH₄ generation. As the increase of doped Ni, this side reaction for CH₄ generation is enhanced. In our experiment, due to the short lifetime, the *CH₃ intermediate is very difficult to detect, while the generation of CH₄ provides an indirect proof to prove the existence of *CH₃ during ethanol dehydration.

Figure 2b. The DME generation rate and product selectivity over Ni_x-W₁₈O₄₉.

Modifications:

- 1) The sentence “Meanwhile, a small part of doped Ni can produce a small amount of hydrogen, and the generated *CH₃ can react with *H for CH₄ byproduct generation as demonstrated in Figure 2b.” was added at p19 of the revised manuscript.

11. In the proposed mechanism, the authors mentioned the occurrence of few intermediates like *CH₃, OH radicals. However, those intermediates were not observed in the in situ FTIR measurement, why?

Response: Thanks for the reviewer’s comment. Our experiment is a gas phase reaction system, and the C-O bond cleavage and C-O-C coupling reactions are rapid. The signals of *CH₃ radical may occupy a position similar to the methyl group (-CH₃) in methanol, making them difficult to distinguish by in-situ FTIR spectroscopy. During methanol dehydration, a tiny amount of CO, CH₄ and H₂ byproducts are detected, and the ratio of CH₄ byproduct is increased from 0.06% to 1.9% as the doped Ni increasing. This result indicates that a small part of Ni sites promotes CH₄ generation. During methanol dehydration, plasmonic hot electrons can facilitate the C-O bond cleavage of methanol on low-valent W sites, generating *CH₃ as intermediates. Meanwhile, *CH₃ can react with *CH₃O on Ni sites to facilitate C-O-C coupling and DME generation. However, a small part of doped Ni can act as active sites for hydrogen generation. A little amount of *CH₃ can react with *H on Ni sites to generate CH₄. As the increase of doped Ni, this side reaction for CH₄ generation is enhanced. Therefore, the generation of CH₄ provides an indirect proof for the existence of *CH₃ radical during methanol dehydration. For *OH intermediate, it is detected in *in-situ* FTIR spectra. The ν_{O-H} vibrational peak intensity shows a progressive increase during the initial 5-10 minutes of light irradiation, suggesting continuous OH radical generation. When the irradiation time is extended to 15 min, the ν_{O-H} vibrational peak intensity decreases, indicating

that the generated OH radicals form water molecules and detach from the catalyst surface.

Figure 6b. In-situ FT-IR spectra of Ni_{0.66}-W₁₈O₄₉ during photocatalytic methanol dehydration reaction under full-spectrum light irradiation.

Modifications:

- 1) The sentence “Meanwhile, a small part of doped Ni can produce a trace amount of hydrogen, and the generated *CH₃ can react with *H for CH₄ byproduct generation as demonstrated in Figure 2b.” was added at p19 of the revised manuscript.
- 2) The sentences “The signal of *OH intermediate was enhanced under full-spectrum light irradiation, and then became weak due to its removal from the catalyst surface as H₂O.” was added at p17 of the revised manuscript.

Reviewer: 3

Comments:

In the submission, the authors constructed Ni-doped plasmonic W₁₈O₄₉ nanowire architecture with exceptional high plasmonic photocatalytic performance in dehydration of methanol to DME and of higher alcohols (>C₂) for alkene. The W₁₈O₄₉ nanowire architecture was reported by the authors as an active plasmonic photocatalyst for the dehydration of alcohols (Ref. 16-18), and in the present work, the authors further screened a series of doped W₁₈O₄₉ nanowire architectures to acquire the best Ni-W₁₈O₄₉ photocatalysts. Although the photocatalytic performance is impressive, the reaction mechanism is rather ambiguous and the comparisons between photocatalytic and thermocatalytic dehydration reactions are not appropriately justified. Thus, I do not consider that the manuscript is suitable for the publication in Nature Communications.

Response: Thanks for the reviewer’s thoughtful and constructive comments. Your suggestions have substantially improved the quality of our manuscript. We have carefully revised the paper in accordance with your comments and hope the revised version can meet the approval of the reviewer.

1. The observation of apparent activation energy for the photocatalytic dehydration of methanol to DME suggested the existence of thermal-activated steps. However, the authors failed to discuss this, which made the proposed reaction mechanism doubtful.

Response: Thank you for reviewer’s comment. The reaction pathway of methanol dehydration

was simulated by DFT calculations as shown in Supplementary Figures 35 and 36. The transition state (TS) for C-O bond cleavage and C-O-C coupling is the key step to determining the kinetics of methanol dehydration for DME formation. Two methanol molecules are adsorbed on the W site and Ni site separately as *methanol(1) and *methanol(2). Then, the O-H bond of *methanol(2) is activated for cleavage to form W-O-H and *CH₃O as intermediates. For the thermocatalysis, *methanol(1) reaction with *CH₃O during TS for C-O bond cleavage and C-O-C coupling to generate DME, while, this process is hard due to the large activation energy (142.4 kJ mol⁻¹). While, upon light irradiation, electrons on low-valent W atoms undergo strong oscillations due to SPR, creating an intense plasmonic field on the surface of Ni-doped W₁₈O₄₉. SPR also excites low-energy electrons to high-energy hot electrons. Both *methanol(1) and *CH₃O are located in the plasmonic field, and plasmonic hot electrons can be transferred to *methanol(1) through the W-O bond, activating and cleaving the C-O bond in methanol for *CH₃ generation and leaving OH adsorbed on W sites as W-O-H. However, *CH₃O connected to Ni prevents hot electron transfer and protects its C-O bond. During TS, the generated *CH₃ from *methanol(1) attacks the oxygen atom of *CH₃O to achieve C-O-C coupling, greatly reducing the activation energy for DME (*DME) formation. The activation energy of photocatalytic methanol dehydration is reduced to 66.4 kJ mol⁻¹ (400 mW cm⁻²). More discussions on the reaction mechanism of methanol dehydration were added in the revised manuscript.

Supplementary Figure 35. DFT calculated energy diagram of methanol dehydration reaction over W₁₈O₄₉ and Ni-doped W₁₈O₄₉.

Supplementary Figure 36. DFT Simulations on reaction pathway of methanol dehydration over Ni-doped W₁₈O₄₉.

Modifications:

- 1) The sentence “According to DFT simulations (Supplementary Figures 35-36), the O-H bond

of *methanol(2) is activated for cleavage to form W-O-H and *CH₃O as intermediates.” was added at p19 of the revised manuscript.

- 2) The sentence “Both *methanol(1) and *methanol(2) are located in the plasmonic field, and plasmonic hot electrons can be transferred to *methanol(1) through the W-O bond, significantly promoting the C-O bond cleavage in methanol for *CH₃ generation and leaving OH adsorbed on W sites as W-O-H.” was modified to be “Both *methanol(1) and *CH₃O are located in the plasmonic field, and the plasmonic hot electrons can be transferred to *methanol(1) through the W-O bond, activating and cleaving the C-O bond in methanol for *CH₃ generation and leaving OH adsorbed on W sites as W-O-H.” at p19 of the revised manuscript.
- 3) The sentence “However, the thermal-activation for C-O cleavage and C-O-C coupling is hard due to the large activation energy (142.4 kJ mol⁻¹) for thermocatalytic methanol dehydration.” was added at p19 of the revised manuscript.
- 4) Figure 6e was updated in the revised manuscript.

Figure 6e. The photocatalytic mechanism of methanol dehydration reaction over Ni_x-W₁₈O₄₉.

- 5) Supplementary Figure 35 and Supplementary Figure 36 were added in the revised Supplementary Materials.

2. Energy diagram of methanol dehydration reaction to DME on the Ni-W₁₈O₄₉ catalyst needs to be calculated.

Response: Thank you for reviewer’s suggestions. We have calculated the energy diagram of the methanol dehydration reaction by DFT simulations, as shown in Supplementary Figure 35. It shows that the Ni-doping in plasmonic W₁₈O₄₉ can reduce the activation energy of methanol dehydration, promoting DME generation.

Supplementary Figure 35. DFT calculated energy diagram of methanol dehydration reaction over W₁₈O₄₉ and Ni-doped W₁₈O₄₉.

Modifications:

- 1) The sentence “DFT simulations further demonstrate that Ni-doping in $W_{18}O_{49}$ can reduce E_a of methanol dehydration as shown in Supplementary Figures 35, boosting DME generation.” at p19 of the revised manuscript.
- 2) The sentences “The Gibbs free energy (ΔG) was calculated using the thermodynamic expression:

$$\Delta G = \Delta E - T\Delta S + \Delta EZPE$$

where $\Delta EZPE$ denotes the zero-point energy correction and ΔS represents the entropy change of intermediate species.

Transition states were located using the climbing image nudged elastic band (CI-NEB) method⁴². The calculations employed 8 intermediate images between initial and final states, with a spring constant of 5.0 eV/\AA^2 and a force convergence threshold of 0.1 eV/\AA .” were added in Methods Section (p23) of the revised manuscript.

- 3) “Henkelman, G. A climbing image nudged elastic band method for finding saddle points and minimum energy paths. *J. Chem. Phys.* **22**, 9901–9904 (2000)” was added as reference 42.
- 4) Supplementary Figure 35 was added in the revised Supplementary Materials.

3. Fig. 1 and Fig. 6e: The authors proposed molecularly-adsorbed CH_3OH as the active species for the dehydration reaction, while in the thermocatalytic reactions, the methoxyl species is the active species (one paper I remembered, *Angew. Chem. Int. Ed.* 2016, 55, 623).

Response: Thank you for the reviewer’s comment. The different mechanisms of thermocatalytic methanol dehydration have been reported in various literatures, such as *ACS Catal.* 2022, 12, 12845; *Appl. Catal. B: Environ.* 2014, 145, 136 and *J. Environ. Chem. Eng.* 2023, 11, 110307. In summary, two kinetic mechanisms, including Langmuir–Hinshelwood (LH) and Eley-Rideal (ER) kinetic expressions, are proposed to describe the reaction procedure over the surface of various catalysts during the thermocatalysis. In the first mechanism (LH), known as dissociative, two methanol molecules are adsorbed on the surface and form methoxy (CH_3O) and OH group species. These species react with each other and produce DME and water. Another proposed mechanism, called associative, follows the ER principle, in which an adsorbed surface methoxy reacts with a molecule not adsorbed on the surface. We also carefully read the reviewer mentioned paper (*Angew. Chem. Int. Ed.* 2016, 55, 623). The authors’ findings demonstrated that methanol readily dissociated at the Ti4c sites of the anatase $TiO_2(001)$ surface to form strongly adsorbed methoxy groups, and the methoxy groups undergo the dehydration coupling reaction to produce DME. This reaction mechanism should be ascribed to the LH kinetic expression, and this important literature has been cited in our revised manuscript (Ref. 39).

In our experiment, we did not observe the clear proofs in Figure 6 (In situ XPS spectra, in situ FT-IR spectra, and methanol-TPD spectra) to support the formation of $*CH_3O$ as intermediates during the methanol hydration reaction. While, from DFT simulations, generated $*CH_3O$ adsorbed on the Ni site (CH_3O-Ni) is favorable for C-O-C coupling with $*methanol(1)$ adsorbed on the W site during the transition state of the methanol dehydration reaction. Our experiment is a gas phase reaction system, meanwhile, the C-O bond cleavage (Figure 6a) and C-O-C coupling

(Figure 6b) reactions are rapidly under light irradiation. The signals of *CH_3 from C-O bond cleavage can not be identified due to the overlapping with methyl group ($-CH_3$) in methanol, making them difficult to distinguish by in-situ FTIR spectroscopy. During photocatalytic methanol dehydration, a trace amount of CO, CH₄ and H₂ byproducts are detected, and the ratio of CH₄ byproduct is increased from 0.06% to 1.9% as the doped Ni increasing. It indicates that a small part of Ni sites promotes CH₄ generation. Plasmonic hot electrons can facilitate the C-O bond cleavage of methanol on low-valent W sites, generating *CH_3 intermediate. Meanwhile, *CH_3 can react with *CH_3O for C-O-C coupling and DME generation, and this process happens rapidly. However, a little amount of *CH_3 can react with the *H on Ni sites for CH₄ generation. The increase of doped Ni enhances this side reaction for CH₄ generation. Therefore, the generation of CH₄ provides an indirect proof to prove the existence of *CH_3 during ethanol dehydration. For *OH intermediate, it is detected in *in-situ* FTIR spectra. The ν_{O-H} vibrational peak intensity shows a progressive increase during the initial 5-10 minutes of light irradiation, suggesting continuous OH radical generation. When the irradiation time is extended to 15 min, the ν_{O-H} vibrational peak intensity decreases, indicating that the generated OH radicals form water molecules and detach from the catalyst surface.

Figure 6a-b. **a** In-situ XPS and **b** In-situ FT-IR spectra of Ni_{0.66}-W₁₈O₄₉ during photocatalytic methanol dehydration reaction under full-spectrum light irradiation.

Modifications:

- 1) The sentence “Meanwhile, a small part of doped Ni can produce a trace amount of hydrogen, and the generated *CH_3 can react with *H for CH₄ byproduct generation as demonstrated in Figure 2b.” was added at p19 of the revised manuscript.
- 2) The sentence “The signal of *OH intermediate was enhanced under full-spectrum light irradiation, and then became weak due to its removal from the catalyst surface as H₂O.” was added at p17 of the revised manuscript.
- 3) The mentioned paper “39. Xiong, F. et al. Methanol conversion into dimethyl ether on the anatase TiO₂(001) Surface. *Angew. Chem. Int. Ed.* **55**, 623–628 (2016).” was cited in the revised manuscript at p28.
4. The authors are encouraged to compare the photocatalytic performance with the typical thermocatalytic performance in the literature. It is also recommended to estimate which temperatures are needed for the Ni-W₁₈O₄₉ catalyst to thermocatalytically catalyzes the CH₃OH dehydration reaction to achieve the photocatalytic performance.

Response: We have compared the photocatalytic performance of plasmonic Ni_{0.66}-W₁₈O₄₉ with the typical thermocatalytic performance of the reported catalysts in the literatures, as shown in Supplementary Table 3. The photocatalytic performance of our catalysts is better than the thermocatalytic performance of most reported optimal catalysts in methanol dehydration for DME generation. Following the reviewer's suggestion, we have determined the temperature required for Ni_{0.66}-W₁₈O₄₉ to exhibit equivalent catalytic performance in thermocatalytic methanol dehydration as observed in the photocatalytic reaction. Specifically, we plotted the Arrhenius plots based on the measured thermocatalytic reaction rates at different temperatures using the Arrhenius equation ($k = Ae^{-Ea/RT}$). By extrapolating the linear fit of $\ln(k)$ versus $1/T$, we determined the temperature required for thermocatalytic methanol dehydration to achieve the same reaction rate as the photocatalytic reaction. The calculation result indicates that a temperature of approximately 277 °C would be required to match the photocatalytic DME generation rate of Ni_{0.66}-W₁₈O₄₉ (136 mmol g⁻¹ h⁻¹).

Supplementary Table 3. Comparison of photocatalytic methanol dehydration performance over plasmonic Ni_{0.66}-W₁₈O₄₉ with the typical thermocatalytic performance in the literatures.

Catalyst	Reaction conditions	Rate (mmol g ⁻¹ h ⁻¹)	Methanol conversion (%)	DME Selectivity (%)
ZSM-5	180 °C, 1.1bar	12.9	23	100 [1]
SAPO-11	280 °C	420	80	100 [2]
Al-modified SBA-15-SO ₃ H	300 °C	53.6	80	100 [3]
6 wt% SiO ₂ /γ - Al ₂ O ₃	300 °C, 1bar	61.3	85	100 [4]
Alumina pillars modified vermiculite	300 °C, 1bar	20.9	80	98 [5]
Ni _{0.66} -W ₁₈ O ₄₉ (This work)	400 mW cm ⁻² , AM1.5 light irradiation, Batch reactor	136	41.5	99%
Ni _{0.66} -W ₁₈ O ₄₉ (This work)	1000 mW cm ⁻² , AM1.5 light irradiation, Batch reactor	784.7	94.9	98.5%
Ni _{0.66} -W ₁₈ O ₄₉ (This work)	Concentrated solar light irradiation (2.3 W cm ⁻²), Continuous reactor	686	92.5	99%

1. Rownaghi, A. A. et al. Selective dehydration of methanol to dimethyl ether on ZSM-5 nanocrystals. *Appl. Catal. B-Environ.* **119**, 56–61 (2012).

- Chen, Z. et al. Fabrication of nano-sized SAPO-11 crystals with enhanced dehydration of methanol to dimethyl ether. *Catal. Commun.* **103**, 1–4 (2018).
- Said, A. E. et al. The catalytic performance of γ -Al₂O₃/red clay as a highly active, selective, and stable catalyst for methanol dehydration to dimethyl ether at competitive low reaction temperature. *Mater. Chem. Phys.* **324**, 129674 (2024)
- Yaripour, F. et al. Catalytic dehydration of methanol to dimethyl ether (DME) over solid-acid catalysts. *Catal. Commun.* **6**, 147–152 (2005).
- Marosz, M. et al. Modified vermiculites as effective catalysts for dehydration of methanol and ethanol. *Catal. Today* **355**, 466–475 (2020).

Modifications:

- The sentence “Compared to the thermocatalytic performance of various reported catalysts shown in Supplementary Table 3, the photocatalytic performance of Ni_{0.66}-W₁₈O₄₉ exhibits a higher DME generation rate.” was added at p8 of the revised manuscript.
- Supplementary Table 3 and literatures was added in the revised Supplementary Materials.

5. The photocatalytic performance varied greatly with the reactant, for example, propanol and isopropanol. Why?

Response: Actually, the dehydration reaction performance of different alcohols is determined by their activation energy barriers (E_a). As displayed in Supplementary Figure 11, we have measured the thermocatalytic dehydration E_a of ethanol, propanol, isopropanol, 1-butanol, isobutanol, 2-butanol, and tert-butanol to be 110, 105, 86.6, 95.8, 89.1, 85.7, and 93.5 kJ mol⁻¹, respectively, over Ni_{0.66}-W₁₈O₄₉ as the catalyst. Isopropanol dehydration reaction has a lower E_a than that of propanol dehydration reaction, indicating a faster kinetics of the isopropanol dehydration process. Under light irradiation, the plasmonic hot electrons on Ni_{0.66}-W₁₈O₄₉ greatly reduce the E_a of alcohol dehydration, boosting alkene generation. The photocatalytic dehydration reaction rates of various alcohols are also determined by their E_a . Under full-spectrum irradiation, the E_a of ethanol, propanol, isopropanol, 1-butanol, isobutanol, 2-butanol, and tert-butanol dehydration reactions is reduced to 39.0, 36.5, 23.2, 27.4, 20.5, 12.0 and 5.4 kJ mol⁻¹, respectively, determining their alkene generation performance to be 32.4, 59.1, 497.6, 290.3, 675.4, 1607.5, and 3694.6 mmol g⁻¹ h⁻¹, respectively.

Supplementary Figure 11. a Arrhenius plots and b the calculated E_a for different alcohol dehydration during thermocatalysis and photocatalysis.

Modifications:

- 1) The sentence “Photocatalytic and thermocatalytic dehydration E_a of different alcohols were measured as shown in Supplementary Figure 11, further confirming the reduced E_a under light irradiation.” was added at p10 of the revised manuscript.

6. How about the carbon balance of the photocatalytic reaction?

Response: Thanks for the reviewer’s suggestion. As the photocatalytic reaction proceeds, the amount of methanol gradually decreases, while the production of DME gradually increases (Supplementary Figure 5). Typically, the production of one DME molecule requires the consumption of two methanol molecules. After one hour of photocatalytic reaction, the system consumes 1020.5 μmol methanol and generates 503.5 μmol DME. The ratio between consumed methanol and generated DME is 2.03, which is close to the theoretical value 2.0. Therefore, the photocatalytic methanol dehydration process maintains a complete carbon balance.

Supplementary Figure 5. The reduction of methanol and generation of DME during photocatalytic reaction.

Modifications:

- 1) The sentence “The molar ratio between consumed methanol and generated DME is 2.03 (Supplementary Figure 5), proving the carbon balance during photocatalytic methanol dehydration reaction.” was added in p5 of the revised manuscript.
- 2) Supplementary Figure 5 was added in the revised Supplementary Materials.

7. Did the authors detected water, the other product for the CH_3OH dehydration reaction?

Response: Thanks for the reviewer’s comment. After the photocatalytic methanol dehydration reaction, visible water droplets formed in the reactor (Supplementary Figure 4a). To precisely determine the liquid reaction products in the reactor, ^1H nuclear magnetic resonance (NMR) measurements were performed. As shown in Supplementary Figure 4a, prior to photocatalysis, the reactor contains only methanol as the liquid phase, but post-reaction ^1H NMR analysis reveals clear water signatures, indicating the generation of water during the photocatalytic process. DME is the main product with a selectivity of 97.6-98.8%, small amounts of H_2 , CH_4 , and CO were detected by gas chromatography-mass spectrometry (GC-MS) during

photocatalytic methanol dehydration over $\text{Ni}_x\text{-W}_{18}\text{O}_{49}$ (Supplementary Figure 3).

Supplementary Figure 4. a Figure of reactor after photocatalytic reaction, b ^1H NMR spectra of liquids before and after photocatalysis.

Supplementary Figure 3. Generation rates of hydrogen, methane and CO during methanol dehydration reaction over $\text{W}_{18}\text{O}_{49}$ and $\text{Ni}_x\text{-W}_{18}\text{O}_{49}$.

Modifications:

- 1) The sentence “ H_2O generation during photocatalysis was confirmed by the ^1H nuclear magnetic resonance spectroscopy (NMR, Supplementary Figure 4)” was added at p5 of the revised manuscript.
- 2) **Supplementary Figure 4** was added in the revised Supplementary Materials.

8. How accurately was the sample temperature measured during the photocatalytic reaction to make the Arrhenius plots?

Response: The sample temperature was measured by the infrared thermal camera (Fluke Tis20+) as described in the experimental section (*The reaction activation energy calculation*), and results have been shown in Supplementary Figure 8 and Supplementary Figure 9. It was considered as an accurate measurement for the sample’s surface temperature during photocatalysis, such as the report by the Hallas group in *Science* (*Science*, 2018, 362, 69).

Sample temperature measurement in the experiment

Supplementary Figure 8. Thermal imaging of the surface temperature of catalyst $\text{Ni}_{0.66}\text{-W}_{18}\text{O}_{49}$ under AM1.5 light irradiation with different light intensities (200, 400, 600, 800 and 1000 mW cm^{-2}).

Supplementary Figure 9. Thermal imaging of the surface temperature of catalyst $\text{Ni}_{0.66}\text{-W}_{18}\text{O}_{49}$ under different light irradiations (UV, Vis, UV-Vis, NIR, Vis-NIR and UV-Vis-NIR, with a fixed intensity of 400 mW cm^{-2}).

9. Please analyze the charge balance mechanism for Ni^{2+} doped $\text{W}_{18}\text{O}_{49}$.

Response: Thanks for the reviewer's good comment. According to the FT-EXAFS spectra analysis, Ni^{2+} was doped into the $\text{W}_{18}\text{O}_{49}$ lattice as single atoms by substituting part of W atoms (W^{6+} , W^{5+} , or W^{4+}) during the one-step solvothermal synthesis. In the synthetic process, when low-valent Ni^{2+} substitutes for high-valent W^{6+} , W^{5+} , or W^{4+} , oxygen vacancies will spontaneously form to maintain charge balance in $\text{Ni}_x\text{-W}_{18}\text{O}_{49}$. Nevertheless, the Ni doping level in the $\text{W}_{18}\text{O}_{49}$ matrix is very low (0.66%), and the increase in oxygen vacancies induced by Ni doping is also minimal. From the Raman spectra shown in Supplementary Figure 14, a new band at 963.8 cm^{-1} is observed in $\text{Ni}_x\text{-W}_{18}\text{O}_{49}$ catalysts, with the enhanced intensity as Ni doping increases, which is assigned to the surface terminal $\text{W}=\text{O}$ stretching mode. This result indicates that the surface-ordered crystal structure of $\text{W}_{18}\text{O}_{49}$ is disrupted by Ni doping, leading to the loss of some lattice oxygen atoms (oxygen vacancies) and the creation of coordinatively unsaturated W atoms on the surface.

Supplementary Figure 14. Raman spectra of samples $W_{18}O_{49}$ and $Ni_x-W_{18}O_{49}$ ($x=0.15, 0.32, 0.66, 1.0,$ and 1.4).

Modifications:

- 1) The sentence “Low-valent Ni^{2+} substitutes for high-valent W^{6+} , W^{5+} , or W^{4+} , oxygen vacancies will spontaneously form to maintain charge balance in $Ni_x-W_{18}O_{49}$.” was added at p11 of the revised manuscript.

10. The information about the light source in Fig. 1 caption was not complete.

Response: Figure 1 is the simulated methanol adsorption on catalysts, maybe the reviewer mentioned is Figure 2. So, we added “Light source: 400 mW cm^{-2} supplied by a 300 W Xe lamp (Perfectlight, PLS-SXE300D) equipped with an AM 1.5G filter.” in the caption of Figure 2.

11. Title: “ $W_{18}O_{49}$ ” should be “ $Ni-W_{18}O_{49}$ ” since $W_{18}O_{49}$ did not have dual active sites.

Response: The title “Semiconductor-Plasmon Drives Dual Active Sites of $W_{18}O_{49}$ for a 590-Fold Boost in Dimethyl Ether Generation” was modified to be “Semiconductor-Plasmon Drives Dual Active Sites of Ni-doped $W_{18}O_{49}$ for Efficient Dimethyl Ether Generation via Methanol Dehydration” at p1 of the revised manuscript.

Manuscript ID: No. NCOMMS-25-19914A

Title: Semiconductor-Plasmon Drives Dual Active Sites of Ni-doped $W_{18}O_{49}$ for Efficient Dimethyl Ether Generation via Methanol Dehydration

Point-by-Point Response to Reviewers

Reviewer: 1

The authors well addressed the reviewers' concerns, the present form can be accepted for publication.

Response: We deeply appreciate the reviewer for his/her time and positive feedback.

Reviewer: 2

The reviewer carefully read the revised manuscript and the responses to reviewers. I appear that the authors have made significant improvements in the revised manuscript by addressing the questions and comments of each reviewer properly. So, I would like to recommend acceptance of the current version of manuscript.

Response: We deeply appreciate the reviewer for his/her time and positive feedback.

Reviewer: 3

Comments:

In the revised submission the authors have improved the manuscript largely. However, the key concerns about the mechanism still hold, particularly the authors failed to rationalize some experimental results using DFT calculations:

Response: We appreciate the reviewer's recognition of our efforts in revising the manuscript. While DFT calculations confirm that Ni-doped $W_{18}O_{49}$ with dual active sites exhibits lower energy barriers for methanol dehydration compared to undoped $W_{18}O_{49}$, these calculations cannot illustrate the role of plasmonic hot electrons in reducing the activation energy (E_a) for DME generation. Therefore, we provide additional FDTD simulations and explanations to response the remaining concerns.

1. Figure 6e illustrates the proposed reaction mechanism. The authors need consider to rationalize the following experimental results: 1) why was NIR light illumination more effective in decreasing the experimentally-measured E_a than UV and Vis light illumination (Fig. 3c); 2) why did the experimentally-measured E_a vary with the power of adopted light?

Response: Thanks for the reviewer's suggestions. As shown in Supplementary Figure 19, the UV-Vis-NIR DRS of $Ni_{0.66}W_{18}O_{49}$ exhibits broad absorption across the full spectral region.

However, the excitation processes differ between the UV and Vis-NIR region, as illustrated in Supplementary Figure 37. Under UV irradiation, electrons on the valance band (VB) of $\text{Ni}_{0.66}\text{-W}_{18}\text{O}_{49}$ are excited across the band gap to the conduction band (CB), generating photoelectrons. Most of these photoelectrons become trapped at the bottom of the CB with low energy, increasing the overall electron density. Meanwhile, a small fraction can transfer to the surface and activate the C-O bond of methanol for cleavage, thereby reducing the E_a . The strong Vis-NIR absorption of $\text{Ni}_{0.66}\text{-W}_{18}\text{O}_{49}$ originates from its intense surface plasmon resonance (SPR) band. Consequently, Vis-NIR irradiation excites SPR, generating hot electrons that activate the methanol C-O bond for cleavage, also resulting in a lower E_a than UV irradiation. As depicted in Supplementary Figure 19, the SPR absorption is significantly stronger in the NIR region than in the Vis region. Therefore, at a fixed intensity of 400 mW cm^{-2} , NIR irradiation generates more hot electrons than Vis irradiation, leading to a lower E_a . The distinct light-matter interactions of $\text{W}_{18}\text{O}_{49}$ in different spectral regions were simulated using FDTD calculations at 350 nm (UV), 650 nm (Vis), and 1200 nm (NIR), as shown in Supplementary Figure 38. Similarly, $\text{Ni}_{0.66}\text{-W}_{18}\text{O}_{49}$ exhibits the strongest interaction with NIR light, enhancing hot electron generation and resulting in the lowest E_a .

Under full-spectrum irradiation, UV-excited photoelectrons become trapped, enhancing the SPR band. This enhancement promotes hot electron generation under concurrent Vis-NIR irradiation, leading to a remarkable reduced E_a for catalytic methanol dehydration. Furthermore, increasing the irradiation power progressively enhances hot electron generation, consequently reducing the E_a in catalytic methanol dehydration.

Supplementary Figure 19. UV-Vis-NIR DRS of $\text{Ni}_{0.66}\text{-W}_{18}\text{O}_{49}$.

Supplementary Figure 37. The excitation processes on $\text{Ni}_{0.66}\text{-W}_{18}\text{O}_{49}$ under UV, Vis-NIR and UV-Vis-NIR light irradiations.

Supplementary Figure 38. Electric field distributions of stacked $\text{W}_{18}\text{O}_{49}$ nanowires at UV (350 nm, a), Vis (650 nm, b), and NIR (1200 nm, c) simulated by FDTD calculations.

Modifications:

- 1) The sentences “The excitations on $\text{Ni}_{0.66}\text{-W}_{18}\text{O}_{49}$ under UV, Vis-NIR and UV-Vis-NIR light irradiation were illustrated in Supplementary Figure 37. Most photoelectrons excited by UV irradiation became trapped at low energy states within conduction band, thereby increasing the electron density.” were added at p19 of the revised manuscript.
- 2) The sentence “Upon light irradiation, electrons on low-valent W atoms undergo strong oscillations due to SPR, creating an intense plasmonic field on the surface of $\text{W}_{18}\text{O}_{49}$.^[18]” was modified to be “Upon Vis-NIR light irradiation, electrons on low-valent W atoms undergo strong oscillations due to SPR, creating an intense plasmonic field on the surface of $\text{W}_{18}\text{O}_{49}$.^[18] (Supplementary Figure 38)” at p19 of the revised manuscript.
- 3) Supplementary Figures 37 and 38 were added in the revised Supplementary Materials.

2. Fig. 6b-c: In addition to the C-O-C region, the C-O region also obviously changed, which needs to be discussed; meanwhile, did the formation of DME give any new features in the C 1s XPS spectra? Moreover, both infrared and XPS results did not demonstrate the high methanol conversions shown in Fig. 3a, why?

Response: Thanks for the reviewer’s suggestions. The *in-situ* FT-IR measurements were conducted in an *in-situ* FT-IR chamber under methanol atmosphere. As shown in Figures 6b and 6c, UV-Vis-NIR irradiation enhanced the peaks in the C–O region, indicating the increased methanol adsorption. UV-Vis-NIR irradiation on $\text{Ni}_{0.66}\text{-W}_{18}\text{O}_{49}$ and $\text{W}_{18}\text{O}_{49}$ promotes

photoelectron trapping, forming more low-valent W^{4+} and W^{5+} species (Figure 4m), which enhances methanol adsorption. Notably, $Ni_{0.66}W_{18}O_{49}$ generates more W^{4+} than $W_{18}O_{49}$ under UV-Vis-NIR irradiation, leading to stronger methanol adsorption. This is consistent with the intensified C-O peak in the FT-IR spectra of $Ni_{0.66}W_{18}O_{49}$ (Figure 6b).

For *in-situ* XPS measurements, samples were methanol-adsorbed, dried, and then placed into the reaction chamber, followed by vacuum pumping before testing. Figure 6a shows a distinct C 1s peak at 286.5 eV (assigned to methanol C-O) on $Ni_{0.66}W_{18}O_{49}$ after methanol adsorption. Note that DME (CH_3-O-CH_3) also exhibits a C-O bond at 286.5 eV, making it impossible to distinguish from the adsorbed methanol in the C 1s XPS spectra. Upon UV-Vis-NIR irradiation, the C-O peak intensity slightly diminishes, suggesting consumption of surface-adsorbed methanol for DME generation.

The purpose of *in-situ* FT-IR and XPS was to detect intermediates and elucidate the methanol dehydration pathway. To prevent rapid intermediate depletion, we used weak light (80 mW cm^{-2}) for these measurements, significantly lower than the light intensity (400 mW cm^{-2}) used in photocatalysis. Consequently, high methanol conversion was not observed spectroscopically. Nevertheless, the strong intermediate peak (C-O-C) assigned to adsorbed *DME was detected in the *in situ* FT-IR spectra for $Ni_{0.66}W_{18}O_{49}$, providing clear evidence that Ni-doping promotes methanol dehydration to DME.

Figure 6b, c. In-situ FT-IR spectra of $Ni_{0.66}W_{18}O_{49}$ (b) and $W_{18}O_{49}$ (c) during photocatalytic methanol dehydration reaction under full-spectrum light irradiation.

Figure 4m. The ratio varies of low-valent W (W^{5+} , W^{4+}) in $W_{18}O_{49}$, $Ni_{0.66}W_{18}O_{49}$, and $Ni_{1.4}W_{18}O_{49}$ after light irradiation.

Figure 6a. In-situ C 1s and W 4f XPS spectra of $\text{Ni}_{0.66}\text{-W}_{18}\text{O}_{49}$ during photocatalytic methanol dehydration reaction.

Modifications:

- 1) The sentence “Full-spectrum light irradiation does not significantly alter these signals. Notably, a new band around 1161.9 cm^{-1} , corresponding to $\nu\text{C-O-C}$, [36, 37] emerges and strengthens with prolonged light irradiation, verifying the methanol dehydration to the DME process.” was modified to “Under full-spectrum light irradiation, a new band around 1161.9 cm^{-1} , corresponding to $\nu\text{C-O-C}$, [36, 37] emerges and strengthens with prolonged light irradiation, verifying the methanol dehydration to the DME process.” at p16 of the revised manuscript.
- 2) The sentence “Furthermore, the $\nu\text{C-O}$ band intensities for both $\text{Ni}_{0.66}\text{-W}_{18}\text{O}_{49}$ and $\text{W}_{18}\text{O}_{49}$ increase with prolonged UV-Vis-NIR irradiation, which is attributed to the light-induced more low-valent W^{4+} and W^{5+} (Figure 4m) for enhanced methanol adsorption.” was added at p17 of the revised manuscript.
- 3) The sentence “The light source used was provided by a xenon lamp coupled through an optical fiber and the light intensity reaching the sample surface was approximately 80 mW cm^{-2} .” was added at p23 of the revised manuscript.
- 4) The sentence “*In-situ* XPS spectra were obtained using an ESCALAB 250Xi+ analyzer, with $\text{Al K}\alpha$ as the excitation source.” was modified to “*In-situ* XPS spectra were obtained using an ESCALAB 250Xi+ analyzer, with $\text{Al K}\alpha$ as the excitation source, and the sample was adsorbed with methanol, dried, and then placed into the reaction chamber, followed by vacuum pumping before testing.” at p24 of the revised manuscript.